# GPS Interferometric Reflectometry measurements of ground surface elevation changes in permafrost areas in northern Canada

Jiahua Zhang[1], Lin Liu[1], Yufeng Hu[2]

[1]Earth System Science Programme, Faculty of Science, The Chinese University of Hong Kong, Hong Kong, 999077, China
[2]College of Geology Engineering and Geomatics, Chang'an University, Xian, 710000, China

*Correspondence to*: Jiahua Zhang (zhangjiahua@link.cuhk.edu.hk)

**Abstract.** Global Positioning System Interferometric Reflectometry (GPS-IR) is a relatively new technique which uses reflected GPS signals to measure surface elevation changes to study frozen ground dynamics. At present, more than 200 GPS stations are operating continuously in the Northern Hemisphere permafrost areas, which were originally designed and maintained for tectonic and ionospheric studies. However, only one site in Barrow, Alaska was assessed to be usable for studying permafrost by GPS-IR. Moreover, GPS-IR has high requirements on ground surface condition, which needs to be open, flat, and homogeneous. In this study, we screen 3 major GPS networks in Canada and identify 12 out of 38 stations located in permafrost areas as useful ones where reliable GPS-IR measurements can be obtained. We focus on the 5 Canadian Active Control System stations and obtain their daily GPS-IR surface elevation changes. We find that the ground surface subsided in Alert, Resolute Bay, and Repulse Bay respectively by $0.61 \pm 0.04$ cm yr$^{-1}$ (2012–2018), $0.70 \pm 0.02$ cm yr$^{-1}$ (2003–2014), and $0.26 \pm 0.05$ cm yr$^{-1}$ (2014–2019). At the other two sites of Baker Lake and Iqaluit, the trends are not statistically significant. The linear trends of deformation were negatively correlated with those of thaw indices in Alert, Resolute Bay, and Repulse Bay. Furthermore, in Resolute Bay, we also find that the end-of-thaw elevations during 2003–2012 were highly negatively correlated with the square root of thaw indices. This study is the first one using multiple GPS stations to study permafrost by GPS-IR. It highlights the multiple useful GPS stations in northern Canada, providing multi-year, continuous, and daily GPS-IR surface deformation, which provide new insights into frozen ground dynamics at various temporal scales and across a broad region.

## 1 Introduction

Since the International Polar Year (2007–2009), permafrost has undergone a warming trend globally, with an average increase of ground temperature at or near the depth of zero annual amplitude by $0.29 \pm 0.12$ °C during 2007–2016 (Biskaborn et al., 2019; Romanovsky et al., 2010; Smith et al., 2010). Warming permafrost causes ground ice melting, active layer thickening, and the release of previously sequestered carbon (Brown et al., 2000; Trucco et al., 2012). It affects hydrological, geomorphological, and biogeochemical processes (Mackey, 1966; Shur and Jorgenson, 2007; Lantuit and Pollard, 2008; Kokelj and Jorgenson, 2013). Measuring and quantifying permafrost changes are crucial for understanding the

dynamics of the active layer and near-surface permafrost (collectively called as frozen ground in this paper), studying the response of permafrost environments to climate change, and assessing the risk of permafrost changes to infrastructures.

Surface elevation changes can serve as an indicator of frozen ground changes. The freeze/thaw of frozen ground is associated with the phase transition of soil moisture, leading to ~9% change of ice volume, due to the density difference between water
and ice. Such volume change in freeze/thaw cycle causes the ground surface to uplift or subside seasonally. Surface deformation can be measured by either traditional benchmark-based methods or modern geodetic and remote sensing ones. The traditional methods use vertical tubes or pipes, anchored deep into the permafrost, as datum references of ground surface for repeat leveling surveys (Mackey, 1983). Modern methods include Interferometric Synthetic Aperture Radar (InSAR), Light Detection and Ranging (LiDAR), and Global Navigation Satellite System (GNSS) positioning. InSAR has been used
to measure and quantify surface subsidence in various permafrost landforms (Liu et al., 2010, 2014, and 2015; Chen et al., 2018). However, InSAR suffers from coarse temporal resolutions and interferometric coherence loss. Furthermore, InSAR measurements need reference points where the surface deformation is known or assumed to be zero. LiDAR has been used to construct differential elevation models to investigate surface deformation (Jones et al., 2015). However, LiDAR surveys are usually conducted at annual or multi-annual intervals. GNSS positioning has also been used to measure and quantify surface
subsidence and uplift (Little et al., 2003; Shiklomanov et al., 2013; Streletskiy et al., 2017). However, those GNSS surveys are usually conducted at the beginning or end of thaw seasons.

Global Positioning System Interferometric Reflectometry (GPS-IR) is a technique which uses reflected GPS signals to measure ground surface changes, such as elevation, soil moisture content, and vegetation growth condition (Larson, 2016,
2019). GPS-IR has been successfully used to study frozen ground dynamics by measuring surface deformation at one station in Barrow, Alaska (Liu and Larson, 2018; Hu et al., 2018). Compared with the aforementioned modern methods, GPS-IR measurements of surface elevation changes have higher temporal resolutions, usually at daily intervals. Their accuracies are on the order of a few centimeters (typically ~2 cm). Their spatial coverage is antenna-height dependent, e.g., 1000 $m^2$ for a 2-m-high antenna. Such spatial coverage fills a gap between regional-scale satellite observations and in situ point
measurements. Furthermore, GPS-IR measurements are free of solid earth movement, such as glacier isostatic adjustment and plate movement (Liu and Larson, 2018). GPS-IR measurements are converted from the vertical distances between the antenna and the reflecting surface. As both the antenna and the surface are equally affected by solid earth movement, GPS-IR measurements can directly reflect frozen ground dynamics.

However, some limitations exist when using GPS-IR. This technique can only be usable in certain surface conditions. It needs the reflecting surface to be open, nearly flat, and relatively homogeneous (Larson, 2016). More than 200 GPS stations are continuously operating in permafrost areas in Northern Hemisphere (Fig. 1). However, only one station, located in Barrow, Alaska has been examined and proved for the use of GPS-IR (Liu and Larson, 2018). The underutilization of the

GPS-IR technique to a large number of existing stations motivates us to assess the usability of all of these sites. We choose Canada as our study area due to the public accessibility of GPS data, abundant weather records, and detailed geological surveys. We first design a three-step framework to identify useful stations under the same protocols and to ensure the reliability of measurements. We then screen all of the major public GPS networks in Canada to identify the usable stations. We then estimate surface deformation by GPS-IR and use them in turn to study frozen ground dynamics.

The significance of this study relies on that it first can provide usable GPS sites in permafrost areas for GPS-IR studies, which are complementary to the existing permafrost monitoring programs such as Circumpolar Active Layer Monitoring (CALM) and Global Terrestrial Network for Permafrost (GTN-P). Permafrost changes have large spatial heterogeneity, due to location, topography, precipitation, and vegetation. Despite the significant increase in the number of in situ sites in the past decades, the CALM and GTN-P sites are still sparse and unevenly distributed. The identified suitable GPS stations can fill in the spatial gaps of the CALM and GTN-P sites. Moreover, GPS-IR measurements are typically continuous and span multiple years. They can be used to study permafrost in a detailed manner and provide new insights into frozen ground dynamics.

In Sect. 2, we describe the mechanism of GPS-IR to measure surface elevation changes, our proposed framework to identify useful GPS stations, and the datasets we used in this study. In Sect. 3, we present basic information of the identified useful GPS stations (e.g., monument material, foundation depth, and antenna height above ground) and environment conditions of the study sites such as biome and surficial material. We then show the results of surface elevation changes in thaw seasons at the study sites in Sect. 4. In Sect. 5, we interpret the GPS-IR results in revealing frozen ground dynamics in various temporal scales, discuss the possible error sources of the results and their limitations in permafrost studies, and present their potential for validating and calibrating space-borne InSAR measurements. We conclude by summarizing the results and findings.

## 2 Methodology

### 2.1 GPS Interferometric Reflectometry (GPS-IR)

Larson (2016 and 2019) presented the principle of GPS-IR and its applications in measuring snow depth, surface soil moisture, vegetation growth condition, and water level. However, the use of GPS-IR for studying permafrost was not explicitly presented. So, here, we describe how GPS-IR retrieves surface elevation changes and their link to ground deformation in permafrost areas.

The input of GPS-IR is signal-to-noise ratio (SNR) data of GPS signals, one of the observables recorded by GPS receivers. It represents the strength of the received signal. SNR series at low satellite elevation angles (e.g., 5°–20° used in this study) oscillate with respect to the elevation angle, due to the interference between direct and reflected signals. The oscillating

frequency mainly depends on the vertical distance between the antenna and the reflecting surface (called reflector height and denoted as $H$). If a GPS station is located above a smooth horizontal surface (e.g., Fig. 2), SNR can be expressed by a sinusoidal function of elevation angle $e$ (Larson, 2016):

$$SNR = A(e) \sin\left(\frac{4\pi H}{\lambda} \sin e + \phi\right), \tag{1}$$

where $A(e)$ is the oscillation amplitude, also varying with $e$; $\lambda$ is the wavelength of the carrier wave of GPS signal; and $\phi$ is the phase. When taking $\sin e$ as an independent variable, the oscillating frequency is:

$$f = \frac{2H}{\lambda}, \tag{2}$$

If $f$ is determined, $H$ can be obtained numerically as

$$H = \frac{f\lambda}{2}, \tag{3}$$


In practice, we divide the SNR series into individual segments corresponding to rising/setting satellite tracks. Then we remove their 2-order polynomial fits and use the residual ones, which are mainly contributed from the reflected signals. For simplicity, we use SNR series hereafter to denote the residual SNR series. We conduct the Lomb-Scargle Periodogram (LSP) analysis on any given SNR series to obtain its frequency spectrum. Then we use the peak value of the spectrum to represent

the frequency $f$ and obtain $H$ by using equation (3). The oversampling parameter of LSP can be determined based on the expected resolution (e.g., 1 mm in this study) of the estimated reflector height. The programs for data processing are available in the software tools of GNSS Interferometric Reflectometry (Roesler and Larson, 2018).

If the monument is deep anchored (e.g., Fig. 2), the GPS antenna is stable with respect to the permafrost. The variation of the

distance $H$ only depends on the change in surface elevation. The change of $H$ is opposite to that of surface elevation, i.e., surface uplift (subsidence) leads to decreasing (increasing) $H$ (Liu and Larson, 2018). For the daily measurements of reflector height, we first assign minus signs to them and then remove the average to represent surface elevation changes.

## 2.2 A framework for identifying useful GPS stations for studying permafrost by GPS-IR

GPS-IR requires the ground surface to be open and relatively flat and smooth. To identify suitable ones from the existing

GPS stations under the same protocols and to ensure the reliability of GPS-IR measurements, we have designed a three-step framework, which is described in detail as follows.

Step 1: Selecting GPS stations in permafrost areas

We first check whether permafrost is present where the GPS station is located. This step aims to identify the GPS stations in

permafrost areas. We use the International Permafrost Association map compiled by Brown et al. (1997), which shows the spatial distribution of permafrost in the Northern Hemisphere.

Step 2: Estimating an azimuth range with an open, flat, and homogeneous ground surface

In this step, we aim to estimate an azimuth range, where the surface is open, nearly flat, and relatively homogeneous at each

selected station in step 1. Normally, ground photos of a GPS station are taken as a part of metadata. In practice, we use ground photos and Google Earth images of a GPS site to check its surrounding environment, then estimate an azimuth range free of obstructions. We also use these images to recognize whether the selected surface is nearly planar and smooth. In Fig. S1, we present ground photos of two GPS stations as typical positive and negative examples, respectively.

Step 3: Ensuring high reliability of GPS-IR measurements

At present, 32 operational GPS satellites orbit around the Earth twice daily. Therefore, multiple SNR series are available within a day. In practice, for any given day, we first process all SNR series within the determined azimuth range and elevation angle range to obtain their $H$ using the standard method summarized in Sect. 2.1. Then, we calculate their median and discard the ones deviating from the median by 0.25 m or more. Then we compute the mean value and the standard

deviation ($\sigma$) of the remaining $H$, and remove those $H$ deviating from the mean value by larger than $3\sigma$ as outliers. The final retained $H$ and their corresponding SNR series are regarded as reliable. We average the final retained $H$ (denote the average as $\bar{H}$) to represent the vertical distance between the antenna and the reflecting surface on that day. The uncertainty of $\bar{H}$ is represented by its standard deviation, i.e., the standard deviation of $H$ divided by the square root of the sample size. To further ensure the reliability of $\bar{H}$, a minimal number of 10 pieces of reliable SNR series are required.

**2.3 Dataset and information**

We use SNR data of L1 C/A signals of the identified GPS sites. L1 C/A is the legacy civilian code broadcasted by all the GPS satellites. By using them, we can obtain GPS-IR measurements spanning over from several years to more than a decade, which enable us to study permafrost in various temporal scales and reveal its response to the changing climate.

To understand and interpret GPS-IR results, we use air temperature and snow depth at the study sites. These measurements are recorded by the nearest weather stations of GPS sites, and can be downloaded from Environment Canada (http://climate.weather.gc.ca/historical_data/search_historic_data_e.html). We also use borehole ground temperatures at the study sites, which are provided by GTN-P (http://gtnpdatabase.org/boreholes).

We also summarize the climate and environment information of the study sites, such as mean annual air/ground temperature (Ednie and Smith, 2015; Environment Canada, http://climate.weather.gc.ca/climate_normals/index_e.html), surficial

material (Cruishank, 1971; Dredge, 1994; Taylor, 1982; Throop et al., 2010), and ground ice content of near-surface permafrost (O'Neill et al., 2019), to provide background information for reference.

## 3 Identified GPS stations and study sites

We have screened all of the three major GPS networks in Canada, namely Canadian Active Control System (CACS), Canadian High Arctic Ionospheric Network (CHAIN), and Portable Observatories for Lithospheric Analysis and Research Investigating Seismicity (POLARIS) (Fig. 3). CACS is a nationwide network and is maintained by the Geodetic Survey Division in conjunction with the Geological Survey of Canada (Lahaye et al., 2001). It serves to build and maintain the Canadian Spatial Reference System, which is fundamental for mapping, navigation, studying crustal deformation. CHAIN

was designed to investigate the impact of solar output on planetary environment (Jayachandran et al., 2009). The network is operated by the University of New Brunswick. It consists of 25 GPS stations, of which three (KUGC, REPC, and QIKC) are shared with CACS. It is important to note that most of the receiver antennas of CHAIN stations are anchored onto the roofs of buildings. Consequently, the monuments may move due to the foundation instability and thermal expansion/contraction of buildings. When using these stations for GPS-IR studies, corrections for such instability should be conducted. POLARIS,

operated by the University of Western Ontario, was initiated for mapping solid Earth's structure and assessing earthquake hazards (Eaton et al., 2005). It includes seven geodetic-quality GPS stations.

Following the framework in Sect. 2.2, we identified 12 GPS stations out of 38 ones located in permafrost areas as suitable ones for GPS-IR studies. Table 1 gives their basic information, including locations, monument types, foundation conditions,

data time spans, and spatial coverages of GPS-IR measurements. The receiver and antenna types are listed in Table S1. Five of them are from CACS, and the rest are from CHAIN. None of the POLARIS stations was identified as suitable.

Given that the GPS-IR measurements of the CHAIN stations might be affected by the unstable buildings, in this study we present and interpret the measured elevation changes at the 5 identified CACS stations. Their monuments are all anchored

into bedrocks (Table 1). Figure 4 shows their ground photos and Fig. S2 shows examples of their SNR series and corresponding LSP spectrum analysis.

These 5 sites are all located in the Canadian Arctic. The climate in this region is dominantly Polar climate due to high latitude. The biomes are mainly tundra and Arctic desert. Permafrost is continuous in this area, and normally its thickness

increases with latitude. In the far north latitude of 75°, permafrost can be thicker than 500 m (Sladen, 2011). Ground temperatures at or near the depth of zero annual amplitude ranged from colder than -15 °C to warmer than -2 °C, and they decreased northward in concert with climate (Smith et al., 2013). During 2008–2014, ground temperatures at the depth of 15 m increased at an average rate of ~0.17 °C yr$^{-1}$ at ten extensively distributed sites in the Canadian Arctic (Ednie and Smith,

2015). Thawing ice-rich permafrost has initiated wide-spread development of thermokarst landforms in this region, such as retrogressive thaw slump (Lantuit and Pollard, 2008; Kokelj et al., 2015) and active layer detachment (Lewkowicz and Harris, 2005; Lewkowicz and Way, 2019).

We summarize the basic regional information of the five sites respectively in Table 2, including biome, land cover, ground ice content of near-surface permafrost, mean annual air temperature (MAAT), and mean annual ground temperature (MAGT). In Alert and Resolute Bay, the biomes are both Arctic Desert due to the high latitude, and the land surfaces are dominantly bare soil. The biomes at the other three sites are all tundra. But, due to their specific locations, the ground surface is mainly bare soil in Repulse Bay, but is covered by a peat layer in Baker Lake, and is sparsely vegetated in Iqaluit.

## 4 Results: surface elevation changes measured by GPS-IR

We obtain multi-year and seasonal time series of surface elevation changes at the 5 CACS sites. In this study, we only present the measurements in thaw seasons, when air temperature is above 0 °C and ground is not covered by snow. The measurements can be found in Zhang et al. (2019, https://doi.pangaea.de/10.1594/PANGAEA.904347).

We build best linear fit to the thaw-season measurements and obtain the trends at the five sites (Fig. 5). We find that in Alert, Resolute Bay, and Repulse Bay, the ground surface subsided at a rate of $0.61 \pm 0.04$ cm yr$^{-1}$ (2012–2018) and $0.70 \pm 0.02$ cm yr$^{-1}$ (2003–2014), and $0.26 \pm 0.05$ cm yr$^{-1}$ (2014–2019), respectively. However, at the other two sites, the displacements of ground surface were $0.04 \pm 0.02$ cm yr$^{-1}$ in Baker Lake during 2010–2017 and $-0.05 \pm 0.02$ cm yr$^{-1}$ in Iqaluit during 2010–2019. These last two trends were not statistically significant (t-test, $\alpha = 0.05$).

In Fig. 6, we present the seasonal surface elevation changes in Resolute Bay. The seasonal results of the other sites can be found in Zhang et al. (2019). During a thaw season, the ground surface typically subsides progressively and reaches its lowest position at the end of season. However, at the Resolute Bay site, the surface elevation changes in the thaw seasons were irregular. The surface uplifted abnormally and significantly within the thaw seasons, for instance in 2003 and 2007. A similar phenomenon was also observed at a site near Yellowknife in Canada by Gruber (2019) using an inclinometer. This phenomenon could be due to the refreezing of soil moisture which migrated from the thawed active layer, or the swelling of soil when it became wet. However, we lack measurements of soil moisture and ice content to investigate the cause of the observed uplift. In addition to such abnormal changes, the elevation changes among the thaw seasons were inconsistent. Given the complexity of these seasonal elevation changes, we turn to investigate the interannual variability and linear trends of surface deformation in Resolute Bay.

## 5 Discussion

In this section, we first interpret GPS-IR measured surface elevation changes in Resolute Bay, as they are the longest among the five sites. We then qualitatively study the linear trends of surface deformation at the five sites. We also discuss the possible error sources of these GPS-IR measurements, the limitations of using GPS-IR measurements in permafrost studies, and their capability in validating and calibrating space-borne InSAR observations.

### 5.1 Interannual variability of end-of-thaw elevations in Resolute Bay

Net seasonal subsidence is an effective indicator of the response of frozen ground to the atmosphere, as it mainly depends on the soil moisture content within the active layer and the heat from the atmosphere. But, as shown in Sect. 4 and Fig. 6, it is challenging to reliably obtain seasonal subsidence in Resolute Bay due to the irregularity and inconsistency of surface elevation changes in thaw seasons. As an alternative, we use the end-of-thaw-season surface elevations to investigate the frozen ground dynamics.


The end-of-thaw elevation is determined as the mean value of the elevations at the last seven days of a thaw season, since the thawing front moves slowly at the end of thaw and the further surface deformation is limited. According to the Stefan equation, active layer thickness is approximately proportional to the square root of thaw index (Brown et al., 2000; Smith et al., 2009; French, 2017). Thaw index is represented by the degree days of thawing (DDT) derived by the accumulation of

daily air temperatures above 0 °C till the end of thaw season. As surface subsidence is mainly caused by ice-melting within the active layer, we compare the end-of-thaw elevations to the square root of the annual thaw indices (Fig. 7).

In Fig 7a, the end-of-thaw-season elevations and $\sqrt{DDT}$ were highly negatively correlated between 2003 and 2012, whereas the end-of-thaw elevations were low with cool summers in 2013 and 2014. To further investigate their correlation, we draw a

scatter plot of end-of-thaw-season elevations versus $\sqrt{DDT}$ (Fig. 7b), but find that the linear line fitted poorly. After removing the measurements in 2013 and 2014, the $R^2$ and Root Mean Square Error (RMSE) of the best linear fit improves significantly, from 0.24 to 0.83 and 2.57 cm to 1.19 cm, respectively (Fig. 7c).

We postulate that the highly negative correlation between the end-of-thaw elevations and $\sqrt{DDT}$ during 2003–2012 was due

to thickening active layer. A larger DDT indicates that more heat is available to penetrate into the deeper part of the frozen ground, leading to active layer thickening, more ice melting within the frozen ground, and thus larger subsidence and lower surface elevation. This assumption of thickening active layer during 2003–2012 is consistent with the borehole ground temperatures during 2008–2012 (Fig. 8). The ground temperatures showed that the thawing front (i.e., the 0 °C isotherm) deepened and exceeded 1 m depth in 2011.


However, in 2013 and 2014, the end-of-thaw elevations were low, even in the relatively cool summers (corresponding to low DDT). This is possibly due to the Markovian behavior of the active layer. Markovian behavior describes the reset of the active layer's response to air temperature after an extremely warm or cold summer, and this new response regime will last till the next extreme thaw season (Nelson et al., 1998). In Resolute Bay, the year 2011 had the warmest summer with the
DDT of 542.9 °C·day, more than 4 times larger than that in 2004 (132.3 °C·day). After 2011, the response of the active layer to the atmospheric forcing may have changed due to the changes in thermal properties of the active layer and ice content at the permafrost table. So, even with low DDT, the maximal thaw depths were still larger than expected, resulting in low end-of-thaw-season surface elevations. Yet, ancillary data such as thermal properties, ice content, soil moisture, and thaw depth are needed to test these postulated changes in the active layer.

**5.2 Linear trends of surface deformation at the CACS sites**

The ground surface deformed differently among the five sites. In this subsection, we study the possible links between linear trends of surface deformation and air temperature, landcover, as well as ground ice near the permafrost table.

We make basic statistics of the annual thaw indices during the study periods at the sites of Alert, Resolute Bay, and Repulse
Bay (Table 3). All of these sites had warming thaw seasons, with trends of 9.35 °C·day yr$^{-1}$ in Alert during 2012–2018 and 8.17 °C·day yr$^{-1}$ in Resolute Bay during 2003–2014, and 66.41 °C·day yr$^{-1}$ in Repulse Bay during 2014–2019, respectively.

The ground surface underwent subsidence with increasing DDT in Alert, Resolute Bay, and Repulse Bay. At these three sites, the surficial materials are sandy soil and barely vegetated (Table 2). Due to the lack of an insulating organic layer, bare
soil facilitates the heat transfer between the atmosphere and the ground. When the climate was warming, the transient layer (i.e., the layer between the active layer and long-term permafrost table and subjected to freeze and thaw seasonally to centennially (Shur et al., 2005)) started to thaw with ground ice melting and surface subsidence, such as that seen in Alert, Resolute Bay, and Repulse Bay, even though they have low ice content in near-surface permafrost (Table 2).

Liu and Larson (2018) obtained surface elevation changes during 2004–2015 at Barrow, Alaska by using GPS-IR, and found a subsidence trend of $0.26 \pm 0.02$ cm yr$^{-1}$. During the same time span, the thaw season in Barrow also had a warming trend with 4.79 °C·day yr$^{-1}$. The results of Liu and Larson (2018) are consistent with ours: warming thaw seasons lead to surface subsidence. These findings in Barrow and our sites indicate that permafrost in high latitudes were degrading and air temperature is the dominant driver.

## 5.3 Possible error sources of GPS-IR-measured surface deformation

GPS-IR measurements of surface elevation changes might be affected by the surrounding environment (e.g., troposphere, vegetation, and soil moisture) and instruments (including antenna and monument). In this section, we discuss the impact of these variables on GPS-IR measurements and their magnitudes.

GPS signals refract when they propagate through the troposphere, leading to changes of propagating velocity and direction. Such refraction effects change the geometry among the direct and reflected signals and the receiver antenna, then introduce bias to reflector height retrievals. Tropospheric bias mainly depends on the antenna height and atmospheric conditions at a given elevation angle (Williams and Nievinski, 2017). In our study, because (1) all sites are located in the Canadian Arctic characterized by a dry and cold climate, and (2) their antenna heights are ~ 2 m (Table 1), the tropospheric biases at these sites are expected to be limited. More quantitatively, we calculate the tropospheric biases at RESO in the thaw season in 2014 by using the astronomical refraction model of Bennett (1982), and present them in Fig. S3. The magnitudes of tropospheric biases are ~1.6 cm and keep relatively steady during the thaw season. As the magnitudes of biases are comparable to the uncertainties of GPS-IR measurements and we focus on the temporal variations of reflector heights, it is not necessary to correct for them.

Soil moisture also affects GPS-IR measurements of surface elevation changes through impacting phases of SNR series. For any given SNR series, soil moisture has slightly different influence on the phase of each point, i.e., $\phi$ in equation (1) is also a function of elevation angle. Taking $\phi$ as a constant in practice introduces bias. Such bias is called compositional reflector height, as it manifests itself by a part of reflector height (Nievinski, 2013). Liu and Larson (2018) simulated the compositional height and found that they are less than 2 cm and their variation range is less than 1 cm, given a variation range of soil moisture between 15% and 40%. In this study, the compositional heights and their variation range are expected to be limited, as the precipitation is light and limited due to the cold and dry polar climate. Moreover, as we focus on the temporal variations of reflector heights at interannual and multi-annual time scales, we expect negligible impact of compositional heights on our results and interpretation.

Regarding the vegetation, at the study sites, the biomes are Arctic desert or tundra. The ground is barely or sparsely vegetated, and the vegetation is short enough, i.e., less than the wavelength of L-band GPS signals. The vegetation is approximately transparent for GPS signals. The impact of vegetation on GPS-IR measurements is therefore negligible.

Antenna gain pattern also impacts GPS-IR measurements. As the GPS stations used in this study were installed originally for geodetic or ionospheric studies, the receiver antennas were designed to favor direct signals with high elevation angles and suppressing signals with low and negative elevation angles, by using asymmetric antenna gain patterns. During the data time

span, the antennas are not replaced. The impacts of antenna gain patterns can be regarded as system biases, and barely impact the GPS-IR results.


As for the monuments of the identified CACS stations, their material is galvanized or stainless steel and aluminum. The coefficients of linear thermal expansion measured at 20 °C of steel and aluminum are $11\sim13 \times 10^{-6}$ m·(m·K)$^{-1}$ and $23.1 \times 10^{-6}$ m·(m·K)$^{-1}$, respectively. Given a temperature variation range of 20 °C in thaw season, for a 2-m-high aluminum/steel monument, the magnitude of thermal expansion is less than 1 mm, at least one order of magnitude smaller than the elevation changes. The thermal expansion/contraction impact is ignorable for GPS-IR measurements.


## 5.4 Limitations of GPS-IR measurements of surface deformation in permafrost studies

GPS-IR measured surface deformation has relatively large uncertainties, whose magnitudes are on the order of a few centimeters (i.e., ~2 cm in Resolute Bay). The uncertainties are mainly caused by the rugged surface, presence of vegetation, and other unexpected disturbances. Such uncertainties make it difficult to study the daily changes of surface elevation based on GPS-IR measurements, and even the seasonal changes if their magnitudes are comparable to those of seasonal subsidence. Resolute Bay is such a case, where daily and seasonal elevation changes cannot be obtained reliably. However, 12-years long measurements enable the interannual variability of end-of-thaw elevations and decadal linear trend to be obtained with high confidence.



Data gaps exist in GPS observations due to instrumental problems. GPS-IR measurements before and after the gaps are contaminated by the bias introduced by the replacement of instruments. The data gaps and bias hinder the study of permafrost with long-term, continuous, and consistent GPS-IR measurements.

The interpretation of GPS-IR measurements in permafrost areas needs ground observations, such as soil temperatures and moistures. However, these data are usually not available at GPS sites, as they were installed initially for tectonic and ionospheric research. Moreover, surface condition records are often brief or absent. This being the case, we usually only have GPS-IR measurements, and lack ancillary data such as ground temperature or soil moisture to help interpret the GPS-IR results.



These limitations indicate that, in the future, better location choices and maintenance of GPS stations and other ground measurement sensors are needed to exploit the full potential of GPS-IR observations in permafrost studies.

## 5.5 Potential of linking GPS-IR measurements to large-scale mapping from InSAR

Both GPS-IR and InSAR can measure surface elevation changes. In Table 4, we summarize their typical temporal and spatial samplings rates, advantages, and limitations. As we mentioned in Sect. 1, GPS-IR measurements are at daily intervals and

local scales. In contrast, space-borne InSAR observations have much coarser temporal resolutions (the shortest to date being 6 days) and larger spatial scales (covering tens of kilometers), and also require a reference point with known surface deformation or assumed stable. These characteristics make GPS-IR and InSAR measurements complementary to each other. GPS-IR measurements could be used to overcome the limitations of InSAR observations. In particular, as GPS-IR measurements are continuously and at daily intervals over a few years to decades, they can provide baseline information for

reference and can validate InSAR observations.

Several major research programs such as Arctic-Boreal Vulnerability Experiment (ABoVE), Next-Generation Ecosystem Experiments (NGEE), and European Space Agency Permafrost Climate Change Initiative (CCI) use remote sensing elevation changes (e.g., InSAR) to investigate permafrost dynamics. GPS-IR measurements can be used to calibrate and

validate them and provide baseline information for historical, current, and future remote sensing measurements from air and space.

## 6 Conclusions

In this study, for the first time, we implement a framework for assessing useful GPS stations for GPS-IR studies in permafrost areas, and identify 12 useful GPS stations extensively distributed across the Canadian permafrost areas. Our

framework can be applied to GPS networks in other regions and nations to identify more usable GPS stations. Our identified useful stations and the potential ones are also complementary to existing monitoring networks such as the CALM and GTN-P programs.

This study is also the first one using multiple GPS stations to study permafrost by GPS-IR. At the 5 identified CACS sites,

we obtain their time series of elevation changes. The ground surface subsided in Alert by $0.61 \pm 0.04$ cm yr$^{-1}$ during 2012–2018, in Resolute Bay by $0.70 \pm 0.02$ cm yr$^{-1}$ during 2003–2014, and in Repulse Bay by $0.26 \pm 0.05$ cm yr$^{-1}$ during 2014–2019. At the other two sites of Baker Lake and Iqaluit, the linear trends are not statistically significant. The trends at Alert, Resolute Bay, and Repulse Bay are negatively correlated to those of annual thaw indices, i.e., warming thaw seasons lead to surface subsidence. This finding indicates that frozen ground at the study sites is sensitive to air temperature changes.


In Resolute Bay, we also find a highly negative correlation between the end-of-thaw elevations and the square-root of thaw indices during 2003–2012 and suspect that it was possibly due to active layer thickening under the warming thaw seasons. And we also find that the end-of-thaw elevations were low even with cool summers in 2013 and 2014. Continuous and daily measurements reveal the complexity of frozen ground dynamics, i.e., the irregularity and inconsistency of seasonal surface

elevation changes and the summer heave in Resolute Bay. To further investigate the dynamics and mechanisms of frozen ground changes, it is important to measure other variables such as ground temperature, soil moisture, and ground ice content.

Our discussion on error sources and limits of GPS-IR measurements recommends that better location choice and maintenance of GPS stations should be conducted to fully use the potential of those stations in frozen ground. The multi-year, continuous, daily GPS-IR measurements with intermediate spatial coverages can validate or calibrate remote sensing observations of elevation changes in permafrost areas.

**Code and data availability**

The software tools of GNSS Interferometric Reflectometry are available from https://www.ngs.noaa.gov/gps-toolbox/GNSS-IR.htm. The SNR observations of CACS GPS sites are available from https://webapp.geod.nrcan.gc.ca/geod/data-donnees/cacs-scca.php?locale=en. The air temperature and snow depth measurements are available from http://climate.weather.gc.ca/historical_data/search_historic_data_e.html. The borehole ground temperature measurements are available from http://gtnpdatabase.org/boreholes. The GPS-IR measurements of surface elevation changes at Alert, Resolute Bay, Repulse Bay, Baker Lake, and Iqaluit in Canada are available from https://doi.pangaea.de/10.1594/PANGAEA.904347.

**Author contribution**

JZ identified the useful GPS stations for GPS-IR studies, conducted data processing to obtain results of surface elevation changes, performed results analysis, and wrote the manuscript. LL helped interpret the results and revise the manuscript. YH guided using the software of GNSS Interferometric Reflectometry and assisted in data processing.

**Competing interest**

The authors declare that they have no conflict of interest.

**Acknowledgments**

We thank CACS, CHAIN, and POLARIS for providing GPS observation files, Environment Canada for weather records, and GTN-P for borehole ground temperatures. We thank Kristine Larson for guidance on GPS-IR and its applications, Sharon Smith for providing thoughtful comments on the interpretation of our results, Michael Craymer for providing ground photos of the GPS station RESO, and Richard Chadwick for the monument conditions of CHAIN stations. This study was supported by the Hong Kong Research Grants Council (CUHK14305618). We thank the two anonymous reviewers and Dr. Felipe Nievinski for their substantially helpful comments.

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

**Table 1. Basic information of the identified GPS stations**

| ID | Site name | GPS network | Lat & Lon (°) | Permafrost zonation | GPS antenna Monument | Monument foundation type and depth | Data time span | Azimuth range used by GPS-IR | Antenna height (m) | Footprint radius (m) |
|---|---|---|---|---|---|---|---|---|---|---|
| ALRT | Alert | CACS | 82.49, -62.34 | Continuous | Galvanized steel pipe | Bedrock, 6 m | 2012–2018 | 270°–360° | 1.9 | 61 |
| RESO | Resolute Bay | | 74.69, -94.89 | Continuous | Steel pipe | Bedrock, 3 m | 2003–2014 | 0°–90° | 2.3 | 67 |
| REPL | Repulse Bay | | 66.52, -86.23 | Continuous | Aluminum pillar | Bedrock, 1.5 m | 2014–2019 | 150°–250° | 2.0 | 61 |
| BAKE | Baker Lake | | 64.32, -96.00 | Continuous | Stainless steel pillar | Bedrock, N.A. | 2010–2017 | 0°–90° | 1.2 | 49 |
| IQAL | Iqaluit | | 63.76, -68.51 | Continuous | Aluminum cylinder | Bedrock, 1 m | 2010–2019 | 30°–120° | 1.7 | 57 |
| PONC | Pond Inlet | CHAIN | 72.69, -77.96 | Continuous | Mounted on buildings | N.A. | 2008–2018 | 150°–240° | 4.5 | 102 |
| HALC | Hall Beach | | 68.77, -81.26 | Continuous | | | 2008–2018 | 180°–360° | 3.7 | 90 |
| IQAC | Iqaluit | | 63.74, -68.54 | Continuous | | | 2008–2018 | 200°–320° | 4.0 | 94 |
| RANC | Rankin Inlet | | 62.82, -92.11 | Continuous | | | 2014–2018 | 300°–150° | 3.8 | 91 |
| FSIC | Fort Simpson | | 61.76, -121.23 | Discontinuous | | | 2014–2018 | 150°–330° | 3.9 | 93 |
| FSMC | Fort Smith | | 60.03, -111.93 | Sporadic | | | 2014–2018 | 30°–120° | 3.9 | 93 |
| SANC | Sanikiluaq | | 56.54, -79.23 | Discontinuous | | | 2008–2018 | 135°–225° | 3.4 | 85 |

**Table 2. Regional background of the study sites**

| | Canadian Forces Station Alert | Resolute Bay | Repulse Bay | Baker Lake | Iqaluit |
|---|---|---|---|---|---|
| Biome | Polar Desert | Polar Desert | Tundra | Tundra | Tundra |
| Landcover[a] | Mainly silts, sands, and shattered rocks filled with ice, ranging from 2.4 to 4 m thick (Taylor, 1982) | Rounded or sub-angular gravels and shelly and fine-grained sands (Cruishank, 1983) | Sands and silts ranging from 1 to 10 m thick (Dredge, 1994) | Coarse gravels and sands with low ice contents underneath a peat layer (Throop et al., 2010) | A thin till veneer with fairly well-developed soil, with sparse vegetation (Throop et al., 2010) |
| Ground ice content of near-surface permafrost[b] | None[e] | Negligible wedge ice and low segregated ice | None | Negligible wedge and segregated ice | Low wedge, segregated, and relict ice |
| MAAT[c] (°C) | -18.0 | -15.7 | -12.1 | -11.8 | -9.8 |
| MAGT[d] (°C) | -11.1 – -14.4 (2007–2011) | -11.9 (2008–2012) | -8.2 (2009–2013) | -7.9 (2006–2007) | -5.6 – -7.1 (2003–2004 & 2011–2012) |

a.  The landcover information is for the areas around the boreholes, which are close to the identified GPS stations.

b.  The ground ice contents are surficial material unit-based, which are simulated with surficial geology, deglaciation, paleo-vegetation, glacial lake and marine limits, and modern permafrost distribution (O'Neill et al., 2019).

c.  MAAT refers to Mean Annual Air Temperature during 1981–2010 (Environment Canada,
http://climate.weather.gc.ca/climate_normals/index_e.html).

d.  MAGT refers to Mean Annual Ground Temperature at or near the depth of zero annual amplitude, except Repulse Bay (Smith et al., 2013). MAGT at Repulse Bay was at the depth of 15 m (Ednie and Smith, 2015).

e.  Note that it is contrary to the field observations (Taylor, 1982) that found ground ice exists in the active layer and near-surface permafrost in Alert.



**Table 3. Basic statistics of annual DDT at Alert, Resolute Bay, and Repulse Bay**

| Site | Data time span | Mean (°C·day) | Trend (°C·day yr$^{-1}$) | Trend of surface deformation[a] (cm yr$^{-1}$) |
|---|---|---|---|---|
| Alert | 2012–2018 | 255.85 | 9.35 | -0.61 ± 0.04 |
| Resolute Bay | 2003–2014 | 319.03 | 8.17 | -0.70 ± 0.02 |
| Repulse Bay | 2014–2019 | 518.63 | 66.41 | -0.26 ± 0.05 |

a. Negative means subsidence, vice versa.

**Table 4. Comparing GPS-IR and space-borne InSAR for measuring surface elevation changes in permafrost areas**

| | GPS-IR | Space-borne InSAR |
|---|---|---|
| Temporal sampling | daily | 6 days to months |
| Spatial coverage | Local, site-specific (about 1000 m$^2$) | Large scale (typically tens to hundreds of kilometers) |
| Need reference of known deformation | No | Yes |
| Advantages | Daily and continuous; Free of reference; Free of solid earth movement | High accuracy (The magnitude of uncertainty is on the order of a few millimeters) |
| Limitations | Surface should be relatively flat and smooth. | Coarse temporal resolution; Loss of coherence; Requiring a reference point. |

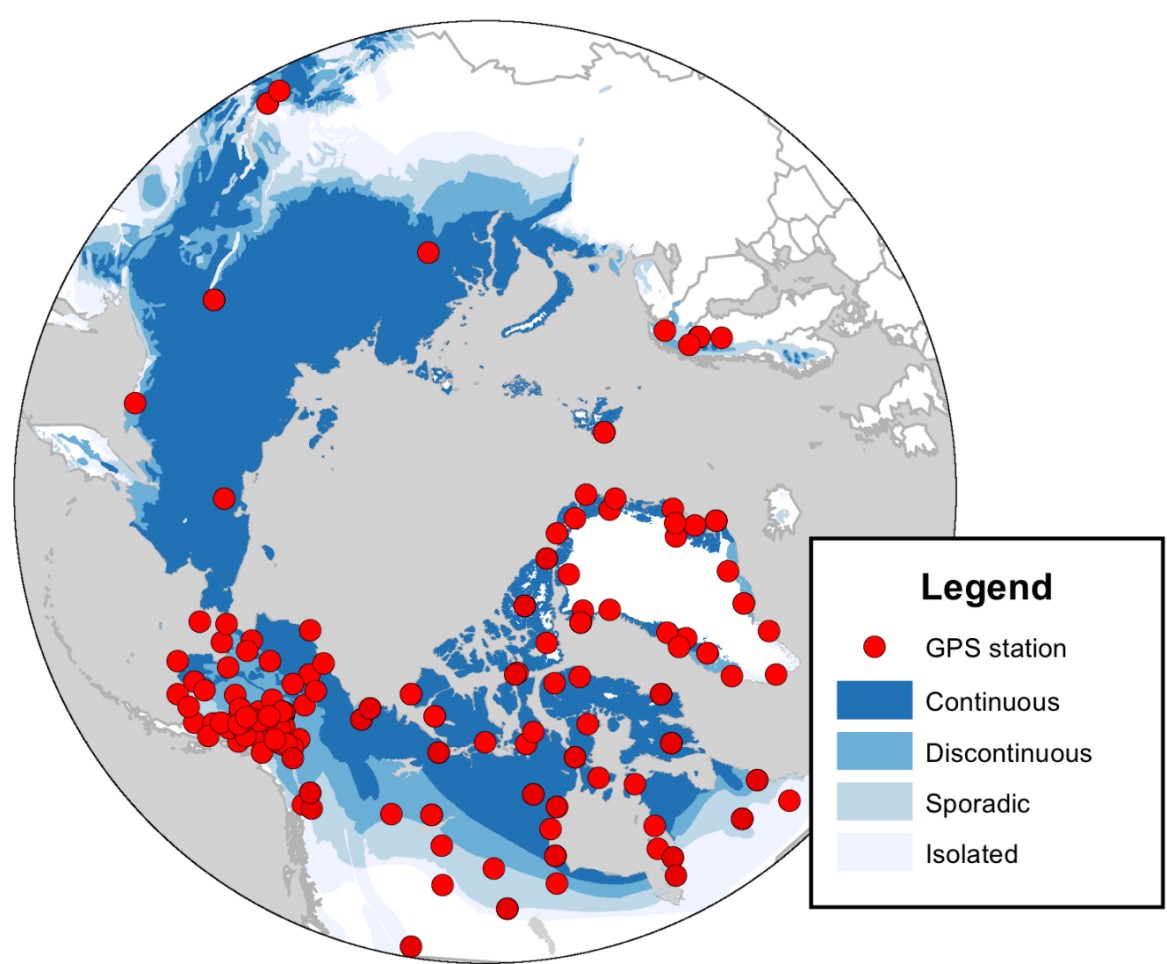

**Figure 1: Locations of continuously operating and open-data GPS stations in the permafrost areas north of 50°N. The permafrost zonation, represented by various colors, is based on Brown et al. (1997).**

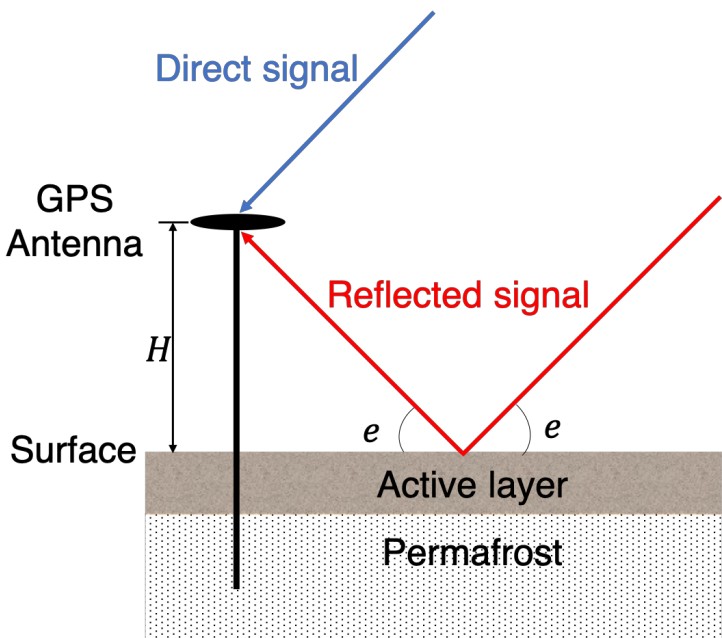

**Figure 2: Schematic diagram showing the geometry of the GPS antenna, GPS signals, and the ground surface in permafrost area.**

*H*, or reflector height, is the vertical distance between the GPS antenna and the surface, and *e* is the satellite elevation angle.

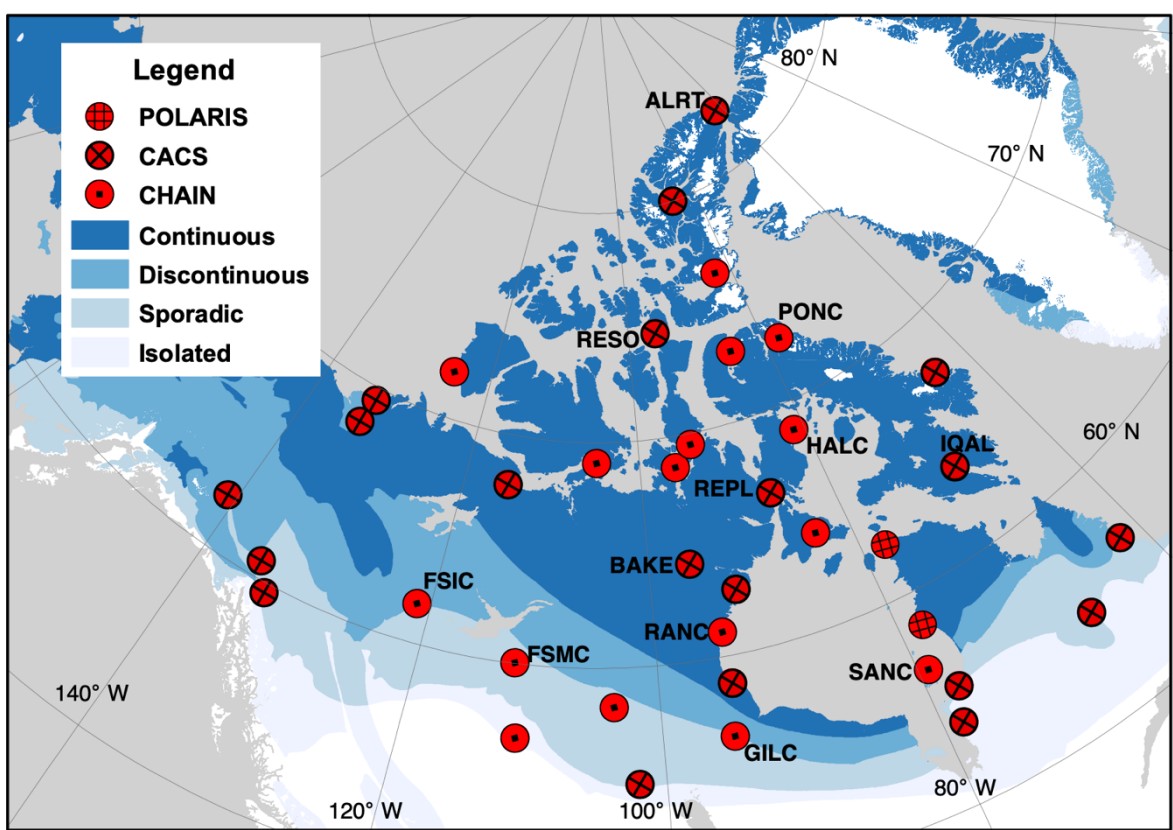

Figure 3: Locations of GPS stations in the Canadian permafrost areas. The identified stations are labeled by their four-character IDs.

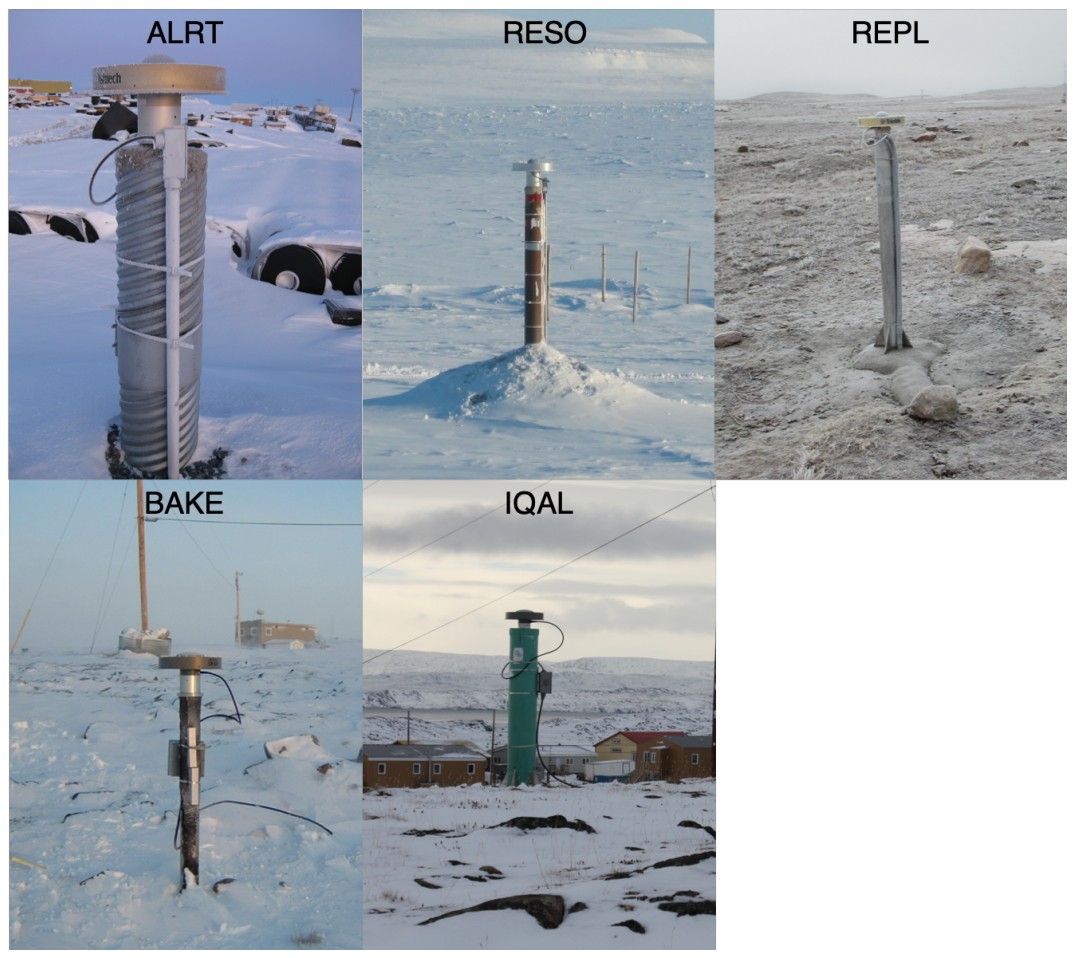

**Figure 4: Ground photos of the identified CACS GPS stations. Source: https://webapp.geod.nrcan.gc.ca/geod/data-donnees/cacs-scca.php?locale=en**

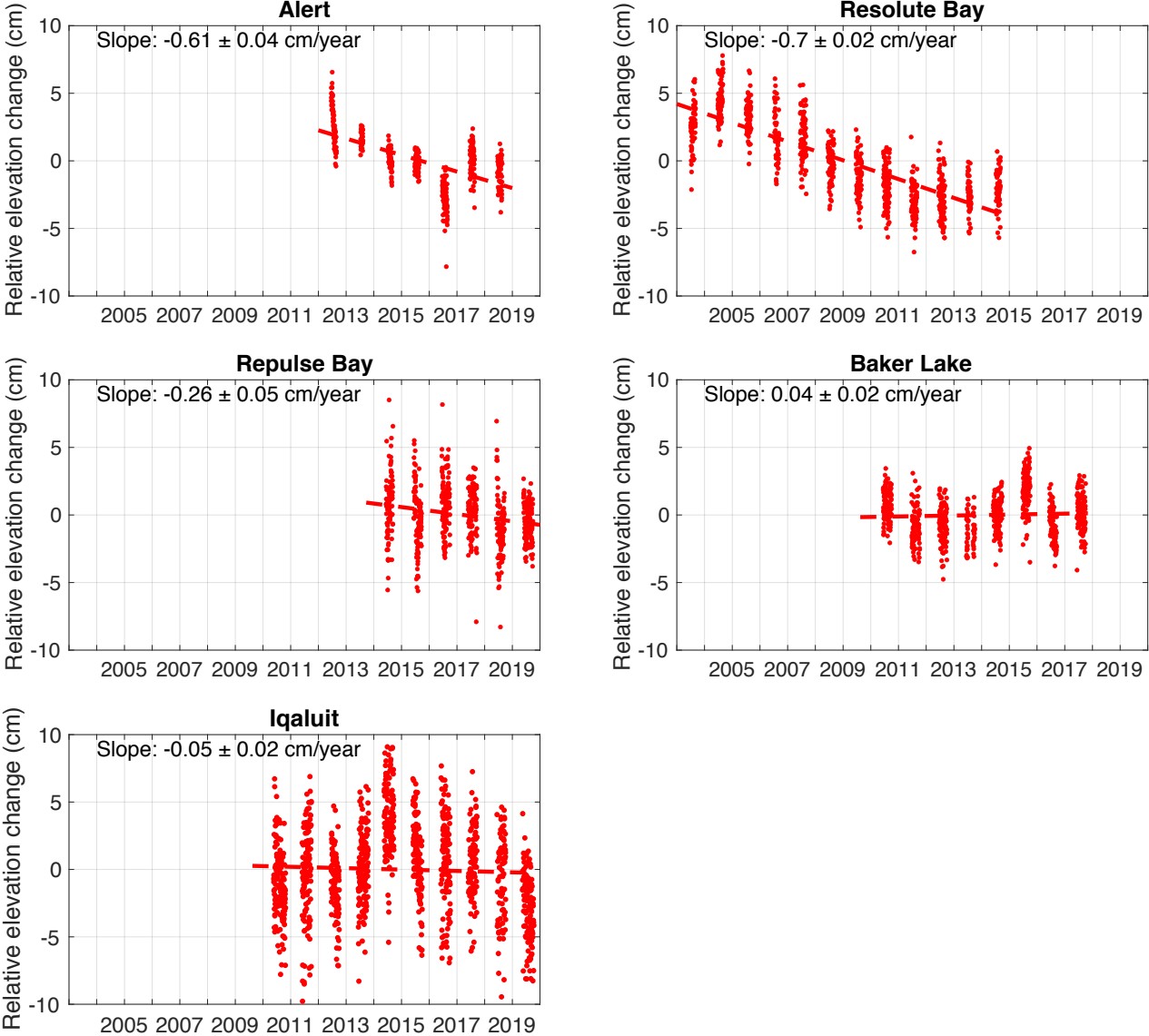

**Figure 5: Time series and the best linear fit dashed lines of surface elevation changes in thaw seasons at the five CACS sites. For clarity, we do not show the error bars. For the y-axis, 'relative' means that the presented elevation changes are referenced to the mean value of the entire records at each site.**


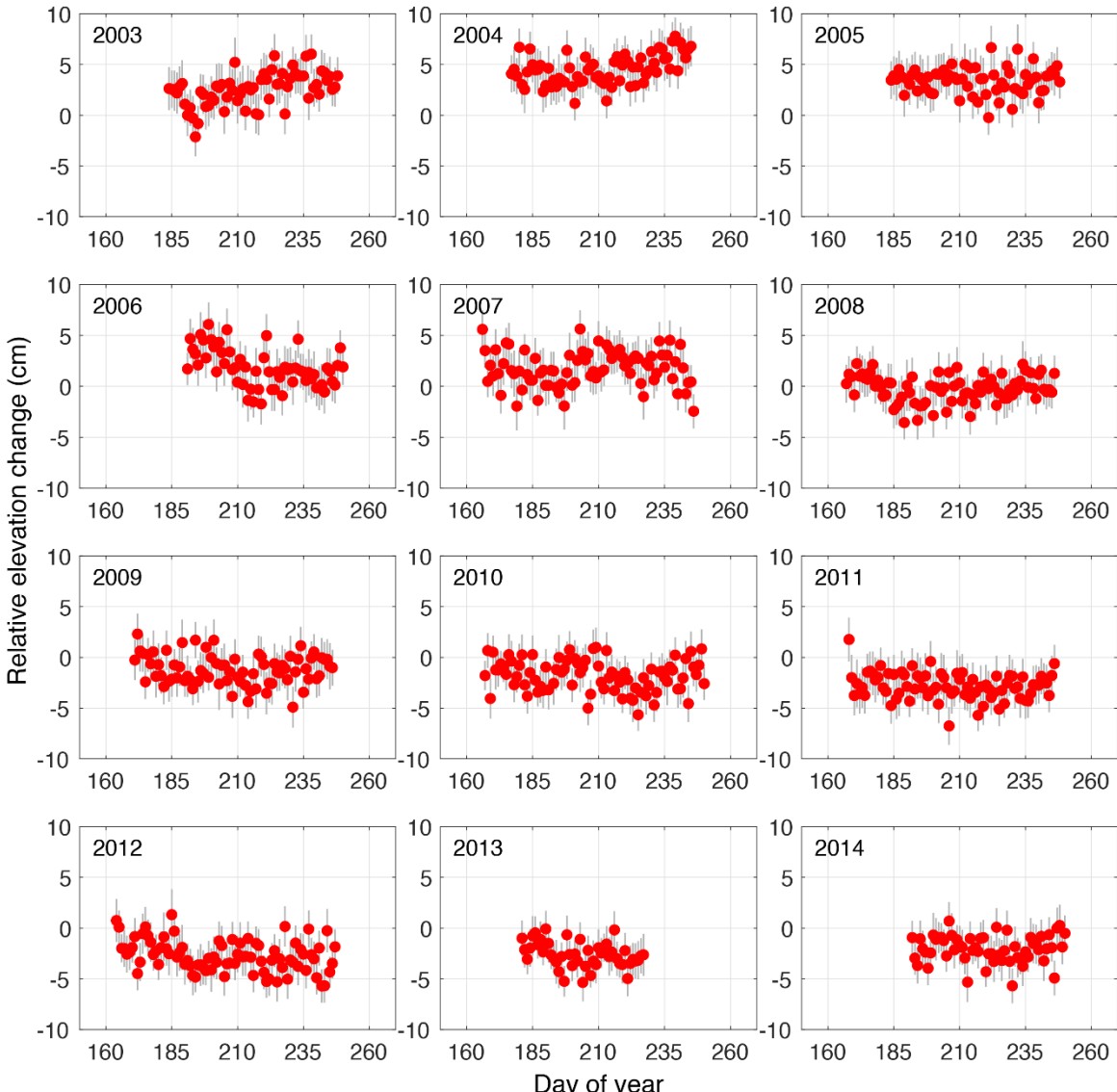

**Figure 6: Surface elevation changes in each thaw season in Resolute Bay during 2003–2014. Red dots denote the measurements in the thaw seasons. Grey error bars denote the uncertainties. The mean value of the measurements has been removed. The shorter thaw season in 2013 was due to the late thawing onset on DOY 181 and early freezing onset on DOY 227, estimated from air temperature and snow depth records.**


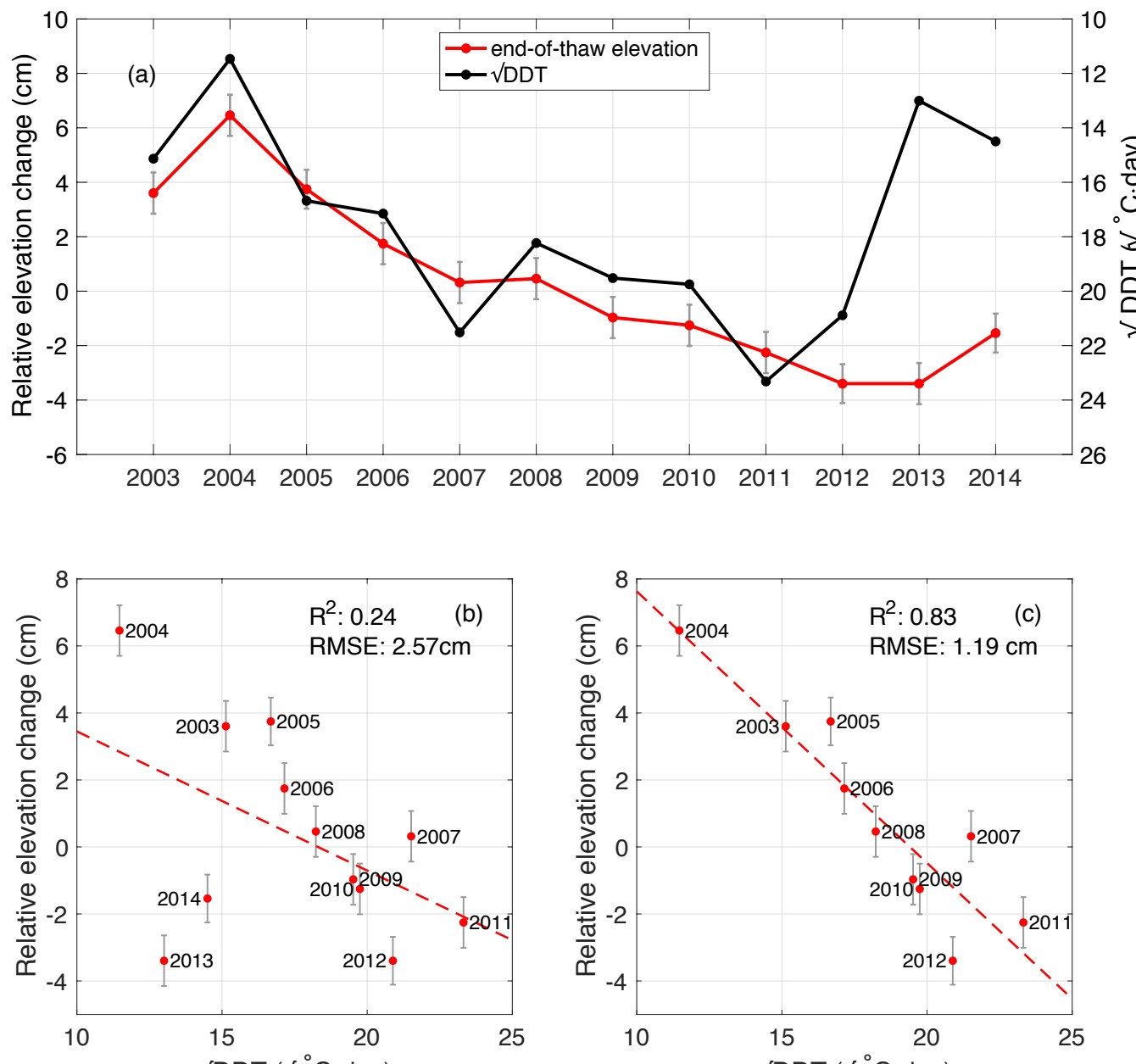

**Figure 7: (a) Time series of the end-of-thaw-season elevations and $\sqrt{DDT}$ during 2003–2014. The right vertical axis for $\sqrt{DDT}$ has been reversed to show the correlation between $\sqrt{DDT}$ and the end-of-thaw elevations. (b) Scatter plots of the end-of-thaw elevations versus $\sqrt{DDT}$. The red dashed line is the best linear fit line. (c) Scatter plot and the best linear fit line of the end-of-thaw-season elevations vs $\sqrt{DDT}$ after removing the measurements of 2013 and 2014.**

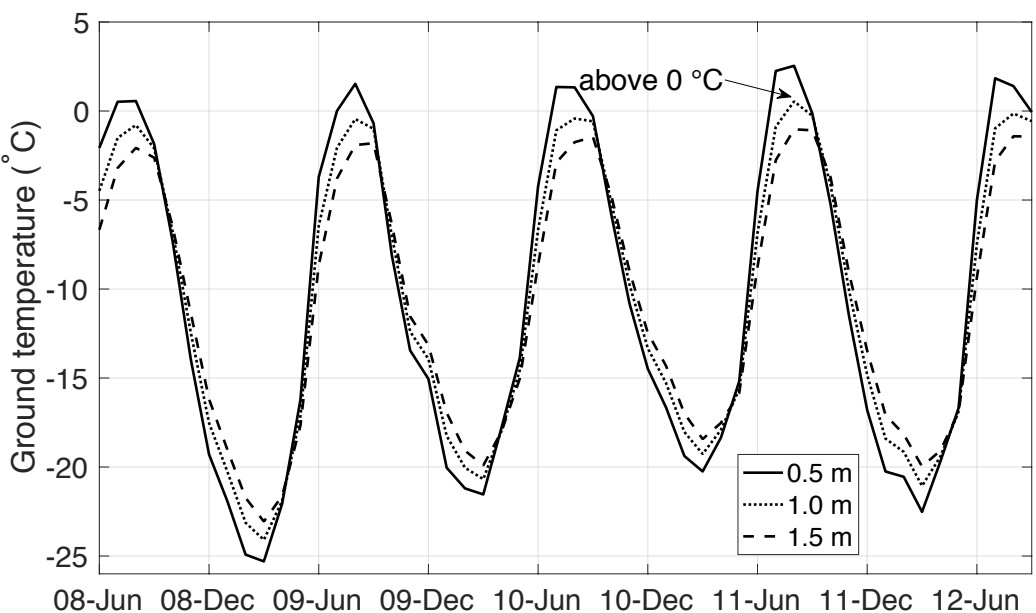

**Figure 8: Time series of monthly ground temperatures at depths of 0.5 m, 1.0 m, and 1.5 m from June 2008 to September 2012 (Ednie and Smith, 2015). In August 2011, the ground temperature at 1.0 m depth was above 0 °C.**