# Peer review of "GPS Interferometric Reflectometry measurements of ground surface elevation changes in permafrost areas in northern Canada"

_The Cryosphere, 2019_

## Referee Comment (RC1) · Anonymous Referee #1 · 15 Oct 2019

GPS-IR is interesting for permafrost studies. New determinations of GPS-IR surface elevation changes can be useful. Unfortunately, I found this manuscript unacceptable and it has to be improved strongly before being published in this journal. The findings of the submitted paper do not add any value to the existing literatures as they are already presented by Liu and Larson (2018) and Hu at al. (2018). Please highlight what is really new and what is the outcome and applicability of this approach in a more prominent way. The following points should be answered: It is already proved by Larson et al. (2008) that the reflector height (H) and the phase of SNR observations are highly correlated. Therefore, some of the H variations should come from the phase variations. SNR observations also are a function of GPS receiving antenna and

[Figure]

GPS signals. The authors have not discussed what types of antennas and signals are investigated, and how their impacts have been moved from the estimated heights. In addition, the penetration depth of microwave signals should be physically estimated to correct the estimated heights. The authors just used low elevation angle observations as SNR oscillations are clearer. However, the tropospheric impact is not negligible for low elevation angle observations even for 2-3 m antenna heights, so its impact should be studied in the paper. The tropospheric refractions seem to have a seasonal impact on GPS-IR (Williams and Nievinski, 2017). The authors have not corrected their GPS-IR solutions. Therefore, the reported slopes are tropospheric-contaminated. In addition, the authors used different antenna monuments (Figure 4), but they did not study the thermal expansion of monuments. The authors used the mean value of each year after the outliers are rejected to estimate and report the slopes, while the results presented in Figure 5 are somehow confusing. For example, at Alert what is the mean for 2016? It seems it is very different from the other 5 years and 2016 cannot represent the reported slope. There are other examples like 2014 at Iqaluit, 2004 and 2011 at Resolute Bay. In addition, it is more complex at Bakes Lake as it seems just a linear fit doesn't represent the GPS station and higher-order polynomials should be used. It would be easier if the median reflector heights for each year were also plotted together with the time series in Figure 5. That would help us to understand if the linear fit is good enough to report surface elevation changes.

---

## Short Comment (SC1) · 22 Dec 2019

The manuscript presents a survey of existing GPS/GNSS stations potentially available for permafrost monitoring via Reflectometry. The application of GPS/GNSS-R for permafrost monitoring seems to be a promising area. The main merit of the present submission is the large number of stations considered. There does not seem to be any methodological innovations. Below I indicate major, moderate, and minor issues for consideration.

——————— MAJOR

[Figure]

The GPS antenna foundation should be discussed in more detail. Currently only the distinction between buildings and steel pipes is given. But not all steel pipes are the same. At least the foundation depth should be given. For example, ALRT is 6-m deep while REPL seems to have concrete slab under the metal pedestal.

——————— MODERATE

The pioneering work of Hu et al. (2018) should be compared and contrasted with the present submission.

More details of the GPS-IR data processing is necessary. For example, what GPS signal was employed – L1-C/A?

Authors should document the GPS receiver and antenna models used in each station, including the time of replacement, at least as supplementary material.

Authors should also acknowledge some of the possible error sources in the simplistic mathematical formulation of eq.(1), where the phase term (phi) is not necessarily constant and can actually vary with soil moisture, vegetation cover, etc.

——————— MINOR

"The uncertainty of H-bar is represented by its standard deviation." -> It should be the standard error of the mean, i.e., the standard deviation divided by the square-root of the sample size.

replace bullet for dot in: °C•day yr-1 -> °CÂůdayÂůyr-1

for a 2-m-height antenna -> for a 2-m-high antenna [or: for 2-m antenna height.]

As the monument is deep anchored (e.g. Fig. 2), the GPS antenna is stable with respect to the permafrost -> If the monument is deep anchored (e.g. Fig. 2), the GPS antenna is stable with respect to the permafrost

---

## Referee Comment (RC2) · Anonymous Referee #2 · 26 Dec 2019

GPS-IR has been used in many terrestrial parameter inverses. The determination of permafrost surface elevation changes via GPS-IR is an interest research. The theory and method of measuring the elevation is the universal method in snow depth and water elevation inverses proposed by Larson. The main work of the manuscript is to filter out a large number of stations that meet the conditions and select the GPS data that can be used in elevation measure. So, this manuscript is unacceptable in its present form and need to add more innovative work. The following questions should be answered:

1. According to the description in the manuscript, the ground of the GPS is buried

deep in the permafrost layer and will not change with the settlement of the active layer. The settlement results obtained are also sub-centimeter level, but the movement of the Earth's plate is vertical to the GPS receiver. The effect of displacement cannot be ignored. It is necessary to consider whether the spatial coordinates of the GPS antenna are constant, so that the ground subsidence can be measured with the GPS antenna as a reference.

2. The photos of the stations showed the surrounding environment are still relatively complicated. The reflected SNR oscillation obtained by this environment should be disordered. The measured reflection SNR sequence diagrams and spectrum analysis results of several stations should be given.

3. The noise of SNR measurement is relatively large, which results in the accuracy of snow thickness measurement with approximate specular reflection being only 0-5cm. The surface of the bare soil is rougher, and the error of height measurement will be larger, but the result is sub-centimeter, how to explain it.

4. The researches of Chew and Small (2014, 2016) showed that vegetation will affect the reflection signal. How to process the influence of vegetation on the reflection signal and height measurement needs to be explained in the manuscript.

---

## Author Comment (AC1) · 22 Jan 2020

**Responses to RC1**

We thank the reviewer for his/her comments. We have addressed all of them in the revised version of our manuscript. Our point-by-point replies are given below. As the reviewer may not be able to read our revised manuscript at this stage of The Cryosphere's review process, the line numbers refer to the previously submitted discussion paper, aiming to point out where the discussion paper has partly addressed the reviewer's comments.

1. Please highlight what is really new and what is the outcome and applicability of this approach in a more prominent way.

**Response**: This comment asks about (1) innovative aspects, (2) outcome and applicability of this approach. Below, we are addressing the innovative aspects of this study in terms of methodology, new findings, and merits of using GPS-IR for studying permafrost. We will respond to the 2nd part of reviewer's comment in a separate thread.

(a) Methodology innovations:

(i) We implemented a framework to identify usable GPS stations from the existing networks to study permafrost by GPS-IR (*Lines 108–133*). To our knowledge, this framework is the first of this kind for permafrost. Previous works by Liu and Larson (2018) and Hu et al. (2018) only used one GPS station (SG27 in Barrow, Alaska). Before our work, it is unknown how many among the more than 200 GPS stations that are operating continuously in the circumpolar permafrost area can be used for GPS-IR. The framework described in this work also serves as a reference for other researchers who would like to screen other GPS stations in other permafrost areas. To our knowledge, the only similar 'data mining' effort for the cryosphere is the PBO H2O project, which lists sites suitable for GPS-IR-based estimates of snow depth (and soil moisture). However, the PBO H2O project ended in 2017 and only archived products are available (https://cires1.colorado.edu/portal/).

(ii) Towards a robust use of GPS-IR, we summarized the limitations of using GPS-IR-estimated surface elevation changes in permafrost studies (*Lines 260–279*). These limitations are related to surrounding environments around stations, instrumentation maintenance, and auxiliary observations such as ground temperature, soil moisture, and ground ice. These limitations indicate that better location choice and maintenance of GPS sites are needed. As

GNSS networks are expanding in the cold regions, the lessons we learned from this study and our recommendations will be helpful for the planning of major geodetic infrastructures while considering the potential applications in permafrost monitoring.

(iii) We considered comprehensively the error sources of GPS-IR-measured surface elevation changes, including tropospheric delay of GPS signals, antenna gain pattern, monument thermal extension/contraction, soil moisture, and vegetation (to be detailed in a new subsection in the revised manuscript).

(iv) We compared the advantages and limitations of GPS-IR measurements and InSAR observations, the latter of which has emerged as a tool for measuring elevation changes over permafrost areas. Our comparisons, detailed in section 5.4 and summarized in Table 4, clearly show that these two methods and their measurements are complementary to each other (*Lines 281–294*). GPS-IR measurements can be used to calibrate and validate InSAR observations and provide baseline information for historical, current, and future remote sensing measurements from air and space.

(b) new findings:

(i) This study is the first one to use multiple GPS sites to study permafrost changes across a broad region by GPS-IR. It is also the first in Canada to use GPS-IR to study permafrost.

(ii) We observed that the surface subsided in Alert during 2012–2017 and Resolute Bay during 2003–2014, in contrary to the uplift in Iqaluit during 2010–2017 (*Lines 185–190*). We found the negative correlation between the linear trends of surface deformation and those of thawing indices (*Lines 251–252*). It indicates that near-surface permafrost is sensitive to air temperature changes and that air temperature is the dominant driver for permafrost stability at these study sites.

(iii) In Resolute Bay, we found that end-of-thaw elevations during 2003–2012 had high negative correlation with the square root of thawing indices. However, in 2013 and 2014, the end-of-thaw elevations were low with cool summers (*Lines 220–224*). This phenomenon is possibly due to the Markovian behaviour of the active layer (*Lines 233–241*), which is worthwhile to be documented and investigated further.

(iv) In Resolute Bay, we observed summer heave of surface in most of thaw seasons during 2003–2014 (*Lines 194–195*). A similar phenomenon is also observed at a different site near Yellowknife in Canada by Gruber (2019) using an inclinometer. These findings reveal that frozen dynamics is rather complex and cannot be all explained by simple Stefan equations.

(c) Merits of using GPS-IR for studying permafrost:

The identified usable GPS sites in this study complement the existing permafrost monitoring programs such as CALM (circumpolar active layer monitoring) and GTN-P (global terrestrial network for permafrost), and provide multi-year, continuous, and daily measurements with intermediate spatial coverages. The changes of permafrost areas have large spatial heterogeneity, due to location, topography, precipitation, and vegetation. Despite of the significant increase of number of in situ sites in the past decades, the CALM and GTN-P sites are still sparse and unevenly distributed. The 12 suitable GPS stations in our study distributed across Northern Canada. Their locations fill in the spatial gaps of the CALM and GTN-P sites (Fig. R1). Moreover, the spatial coverage of GPS-IR at one site is on the order of 1000 $m^2$, nicely bridging point observations and regional-scale remote sensing measurements (*Lines 54–56*). Moreover, numerous GPS stations, e.g., RESO in resolute Bay, have been in operation for more than a decade. They can provide long-term, continuous, and daily measurements, which aid in studying permafrost in a detailed manner and provide new insights to permafrost dynamics.

[Figure]

Figure R1: Locations of the identified GPS stations in Northern Canada and the CALM sites and GTN-P boreholes. The IDs of GPS stations are presented. The types of permafrost distribution are indicated by various colours.

1. Please highlight what is really new and what is the outcome and applicability of this approach in a more prominent way.

**Response**: Corresponding to the innovations presented above, the major outcomes of this approach in this study are (1) a framework guiding to filter out suitable GPS stations in permafrost areas, where GPS-IR is feasible, (2) 12 identified suitable GPS stations distributed across Northern Canada, (3) Multi-year, continuous, and daily measurements of surface deformation at five geodetic-quality GPS sites, (4) new findings in the dynamics of the active layer and permafrost.

Applicability of GPS-IR depends on the surface conditions, which should be relatively horizontal and smooth. This is the reason why we proposed a framework to identify suitable

ones from existing stations. We also summarize the limitations of GPS-IR measurements in permafrost studies. The applicability and limitations of GPS-IR indicate that better location choices and instrument maintenance should be made in the future to fully realize the potentials of the GPS stations installed in frozen areas.

2. It is already proved by Larson et al. (2008) that the reflector height (H) and the phase of SNR observations are highly correlated. Therefore, some of the H variations should come from the phase variations.

**Response**: We do not think Larson et al. (2008) proved the high correlation between reflector height and the phase of SNR series. Instead, their study found a high correlation between the phase of SNR series and surface soil moisture. To explain this finding, they introduced the term of apparent reflector height, which is converted from the phase (equation (1) can be rewritten as $SNR = A(e) \sin(\frac{4\pi}{\lambda}\left(H + \frac{\lambda\phi}{4\pi \sin e}\right)\sin e)$). They thought that soil moisture affects apparent reflector height (i.e., "When the soil is wet, the apparent reflector is close to the surface; as it dries, the reflection depth is several cm deeper" in Larson et al. (2008)), then affects phase. Apparent reflector height (converted from the phase) cannot be mixed up with the reflector height (converted from the frequency).

3. SNR observations also are a function of GPS receiving antenna and GPS signals. The authors have not discussed what types of antennas and signals are investigated, and how their impacts have been moved from the estimated heights.

**Response**: In this study, we used SNR data of GPS L1 C/A signals, as L1 C/A is the legacy civilian signal broadcasted by all satellites and long-term measurements can be obtained. As for the antenna, its gain pattern impacts GPS-IR measurements. The GPS stations used in this study were originally installed and maintained for geodetic or ionospheric studies. Their antennas were designed to favour direct signals with high elevation angles and suppressing signals with low and negative elevation angles, by using asymmetric antenna gain patterns. During the data time spans, antennas were not replaced. The impact of antenna gain pattern can be regarded as a system bias, and barely impact the GPS-IR results.

In the revised manuscript, we have explicitly presented the antenna types and included more details of data processing. And we also have added a subsection discussing possible error sources of GPS-IR measurements.

4. In addition, the penetration depth of microwave signals should be physically estimated to correct the estimated heights.

**Response**: The penetration depth of microwave signals, i.e., the depth where the power of signal reduces to 1/e of its value at soil surface, is not appropriate for defining the depth of reflector (Chew et al., 2014). The signal might penetrate into soil to some depth, but it does not mean that a significant part of the signal would reflect back from that depth to antenna. The interaction between signal and ground surface is rather complex. Comparing to penetration depth, it is more important to consider how changes of ground permittivity would affect SNR data. Zavorotny et al. (2010) built a forward model to simulate SNR, with consideration of antenna gain pattern and surface reflectivity however no penetration depth. The simulated SNR and their GPS-IR retrievals have a good agreement with the experimental ones. This simulation indicates that reflected signal from the penetration depth has a minor contribution to SNR. Furthermore, we focus on the temporal variations of reflector heights. We do not expect significant temporal changes in penetration depth during the data time spans. Therefore, we do not see a need to estimate penetration depth and correct for it.

5. The authors just used low elevation angle observations as SNR oscillations are clearer. However, the tropospheric impact is not negligible for low elevation angle observations even for 2-3 m antenna heights, so its impact should be studied in the paper. The tropospheric refractions seem to have a seasonal impact on GPS-IR (Williams and Nievinski, 2017). The authors have not corrected their GPS-IR solutions. Therefore, the reported slopes are tropospheric-contaminated.

**Response**: Tropospheric refraction may introduce biases in the estimated reflector height, which, however, barely affect the retrieved temporal elevation changes as presented in this study. The tropospheric bias mainly depends on antenna height and atmospheric conditions at a given satellite elevation angle (Williams and Nievinski, 2017). The GPS stations used in this study are located in the Canadian Arctic, where the climate is dry and cold. The antenna heights of these stations are ~2 m. Conceptually, the tropospheric bias of GPS-IR measurements of reflector height is limited.

We calculate the tropospheric bias by using the astronomical refraction model of Bennett (1982) and SNR data in Resolute Bay in the thaw season of 2014. The results show that the mean bias is 1.6 cm, and that they are relatively steady with a variation range of merely 3 mm

(Fig. R2(a)). The biases do not show significant seasonal pattern. And their magnitudes are comparable to the uncertainties of our GPS-IR measurements (Fig. R2(b)). As we focus on the temporal variations of reflector heights, instead of their absolute values, it is unnecessary to correct for the tropospheric biases.

[Figure]

Figure R2: (a) Time series of tropospheric biases in estimated reflector heights in Resolute Bay in the thaw season (i.e., DOY 192–250) of 2014. They are the mean values of the tropospheric biases of all satellite tracks; their standard deviations are indicated by error bars. (b) Time series of GPS-IR-estimated surface elevation changes shown in Fig. 6 in the submitted discussion paper and their 7-day moving averages.

6. In addition, the authors used different antenna monuments (Figure 4), but they did not study the thermal expansion of monuments.

**Response**: The monuments of the 5 identified CACS stations are made of aluminium or Galvanized/Stainless steel, whose linear thermal expansion/contraction coefficients are $11\sim13 \times 10^{-6}$ and $23.1 \times 10^{-6}$ m/(m K), respectively. Given a temperature variation range of 20 °C in thaw season, for a 2-m-high aluminum/steel monument, the magnitude of thermal

expansion is less than 1 mm, at least one order of magnitude smaller than the elevation changes. The thermal expansion/contraction impact is ignorable for GPS-IR measurements.

7. The authors used the mean value of each year after the outliers are rejected to estimate and report the slopes, while the results presented in Figure 5 are somehow confusing. For example, at Alert what is the mean for 2016? It seems it is very different from the other 5 years and 2016 cannot represent the reported slope. There are other examples like 2014 at Iqaluit, 2004 and 2011 at Resolute Bay.

**Response**: Sorry about the confusion caused. We would like to clarify that we use all daily GPS-IR measurements in thaw seasons during the entire data time span to obtain the linear trend at each site. For example, in Alert, the linear trend corresponds to the daily measurements in thaw seasons during 2012–2017. We do not use the mean value of each year or reject any outliers.

The deviation of the measurements of Alert in 2016 (or those of Iqaluit in 2014 and those of Resolute Bay in 2004 and 2011) from the best linear fit is due to the interannual variability of changes in the active layer and near-surface permafrost. Such interannual variability is related to environmental variables including precipitation, soil temperature, soil moisture content, and ground ice condition. However, it is challenging to study the interannual variability as most of these ancillary records are not available. This is one of the limitations of using GPS-IR measurements to study permafrost as we discussed in section 5.3 (*Lines 272–276*).

8. In addition, it is more complex at Bakes Lake as it seems just a linear fit doesn't represent the GPS station and higher-order polynomials should be used. It would be easier if the median reflector heights for each year were also plotted together with the time series in Figure 5. That would help us to understand if the linear fit is good enough to report surface elevation changes.

**Response**: The linear trend is superimposed by seasonal variation, interannual variability, and possibly sub-decadal pattern. Given the data time span, it is not justified to simply use a high-order polynomial to fit the time series, as it is not long enough. The GPS site in Baker Lake is still operating. It would be clearer whether a sub-decadal pattern exists when we have at least decade-long measurements.

We tend not to use median reflector heights of each year. The time series of measurements is a combination of linear trend, interannual variability, seasonal variation, and possible sub-decadal pattern. It cannot be represented by the time series of median reflector heights.

Reference:

Bennett, G. G. (1982). The Calculation of Astronomical Refraction in Marine Navigation. Journal of Navigation, 35(2), 255–259. https://doi.org/DOI:10.1017/S0373463300022037

Chew, C. C., Small, E. E., Larson, K. M., & Zavorotny, V. U. (2014). Effects of near-surface soil moisture on GPS SNR data: Development of a retrieval algorithm for soil moisture. IEEE Transactions on Geoscience and Remote Sensing, 52(1), 537–543. https://doi.org/10.1109/TGRS.2013.2242332

Gruber, S. (2019). Ground subsidence and heave over permafrost: hourly time series reveal inter-annual, seasonal and shorter-term movement caused by freezing, thawing and water movement. The Cryosphere Discussions, 2019, 1–18. https://doi.org/10.5194/tc-2019-227

Hu, Y., Liu, L., Larson, K. M., Schaefer, K. M., Zhang, J., & Yao, Y. (2018). GPS Interferometric Reflectometry Reveals Cyclic Elevation Changes in Thaw and Freezing Seasons in a Permafrost Area (Barrow, Alaska). Geophysical Research Letters, 45(11), 5581–5589. https://doi.org/10.1029/2018GL077960

Larson, K. M., Small, E. E., Gutmann, E. D., Bilich, A. L., Braun, J. J., & Zavorotny, V. U. (2008). Use of GPS receivers as a soil moisture network for water cycle studies. Geophysical Research Letters, 35(24), 1–5. https://doi.org/10.1029/2008GL036013

Liu, L., & Larson, M. (2018). Decadal changes of surface elevation over permafrost area estimated using reflected GPS signals. The Cryosphere, 12(2), 477–489. https://doi.org/10.5194/tc-12-477-2018

Williams, S. D. P., & Nievinski, F. G. (2017). Tropospheric delays in ground- based GNSS multipath reflectometry—Experimental evidence from coastal sites. Journal of Geophysical Research: Solid Earth, 122(3), 2310–2327. https://doi.org/10.1002/2016JB013612

Zavorotny, V. U., Larson, K. M., Braun, J. J., Small, E. E., Gutmann, E. D., & Bilich, A. L. (2010). A Physical Model for GPS Multipath Caused by Land Reflections: Toward Bare Soil Moisture Retrievals. IEEE Journal of Selected Topics in Applied Earth Observations and Remote Sensing, 3(1), 100–110. https://doi.org/10.1109/JSTARS.2009.2033608

---

## Author Comment (AC2) · 22 Jan 2020

**Responses to RC2**

We thank the reviewer for his/her insightful and constructive comments. We have addressed all of them in the revised version of our manuscript. Our point-by-point replies are given below. As the reviewer may not be able to read our revised manuscript at this stage of The Cryosphere's review process, the line numbers refer to the previously submitted discussion paper, aiming to point out where the discussion paper has partly addressed the reviewer's comments.

GPS-IR has been used in many terrestrial parameter inverses. The determination of permafrost surface elevation changes via GPS-IR is an interest research. The theory and method of measuring the elevation is the universal method in snow depth and water elevation inverses proposed by Larson. The main work of the manuscript is to filter out a large number of stations that meet the conditions and select the GPS data that can be used in elevation measure. So, this manuscript is unacceptable in its present form and need to add more innovative work.

**Response**: The main work is not limited to the 12 identified useful GPS stations. Below, we present the innovative aspects of this study in terms of methodology, new findings, and merits of using GPS-IR for studying permafrost.

(a) Methodology innovations:

(i) We implemented a framework to identify usable GPS stations from the existing networks to study permafrost by GPS-IR (*Lines 108–133*). To our knowledge, this framework is the first of this kind for permafrost. Previous works by Liu and Larson (2018) and Hu et al. (2018) only used one GPS station (SG27 in Barrow, Alaska). Before our work, it is unknown how many among the more than 200 GPS stations that are operating continuously in the circumpolar permafrost area can be used for GPS-IR. The framework described in this work also serves as a reference for other researchers who would like to screen other GPS stations in other permafrost areas. To our knowledge, the only similar 'data mining' effort for the cryosphere is the PBO H2O project, which lists sites suitable for GPS-IR-based estimates of snow depth (and soil moisture). However, the PBO H2O project ended in 2017 and only archived products are available (https://cires1.colorado.edu/portal/).

(ii) Towards a robust use of GPS-IR, we summarized the limitations of using GPS-IR-estimated surface elevation changes in permafrost studies (*Lines 260–279*). These limitations are related to surrounding environments around stations, instrumentation maintenance, and auxiliary observations such as ground temperature, soil moisture, and ground ice. These limitations indicate that better location choice and maintenance of GPS sites are needed. As GNSS networks are expanding in the cold regions, the lessons we learned from this study and our recommendations will be helpful for the planning of major geodetic infrastructures while considering the potential applications in permafrost monitoring.

(iii) We considered comprehensively the error sources of GPS-IR-measured surface elevation changes, including tropospheric delay of GPS signals, antenna gain pattern, monument thermal extension/contraction, soil moisture, and vegetation (to be detailed in a new subsection in the revised manuscript).

(iv) We compared the advantages and limitations of GPS-IR measurements and InSAR observations, the latter of which has emerged as a tool for measuring elevation changes over permafrost areas. Our comparisons, detailed in section 5.4 and summarized in Table 4, clearly show that these two methods and their measurements are complementary to each other (*Lines 281–294*). GPS-IR measurements can be used to calibrate and validate InSAR observations and provide baseline information for historical, current, and future remote sensing measurements from air and space.

(b) new findings:

(i) This study is the first one to use multiple GPS sites to study permafrost changes across a broad region by GPS-IR. It is also the first in Canada to use GPS-IR to study permafrost.

(ii) We observed that the surface subsided in Alert during 2012–2017 and Resolute Bay during 2003–2014, in contrary to the uplift in Iqaluit during 2010–2017 (*Lines 185–190*). We found the negative correlation between the linear trends of surface deformation and those of thawing indices (*Lines 251–252*). It indicates that near-surface permafrost is sensitive to air temperature changes and that air temperature is the dominant driver for permafrost stability at these study sites.

(iii) In Resolute Bay, we found that end-of-thaw elevations during 2003–2012 had high negative correlation with the square root of thawing indices. However, in 2013 and 2014, the end-of-thaw elevations were low with cool summers (*Lines 220–224*). This phenomenon is possibly due to the Markovian behaviour of the active layer (*Lines 233–241*), which is worthwhile to be documented and investigated further.

(iv) In Resolute Bay, we observed summer heave of surface in most of thaw seasons during 2003–2014 (*Lines 194–195*). A similar phenomenon is also observed at a different site near Yellowknife in Canada by Gruber (2019) using an inclinometer. These findings reveal that frozen dynamics is rather complex and cannot be all explained by simple Stefan equations.

(c) Merits of using GPS-IR for studying permafrost:

The identified usable GPS sites in this study complement the existing permafrost monitoring programs such as CALM (circumpolar active layer monitoring) and GTN-P (global terrestrial network for permafrost), and provide multi-year, continuous, and daily measurements with intermediate spatial coverages. The changes of permafrost areas have large spatial heterogeneity, due to location, topography, precipitation, and vegetation. Despite of the significant increase of number of in situ sites in the past decades, the CALM and GTN-P sites are still sparse and unevenly distributed. The 12 suitable GPS stations in our study distributed across Northern Canada. Their locations fill in the spatial gaps of the CALM and GTN-P sites (Fig. R1). Moreover, the spatial coverage of GPS-IR at one site is on the order of 1000 $m^2$, nicely bridging point observations and regional-scale remote sensing measurements (*Lines 54–56*). Moreover, numerous GPS stations, e.g., RESO in resolute Bay, have been in operation for more than a decade. They can provide long-term, continuous, and daily measurements, which aid in studying permafrost in a detailed manner and provide new insights to permafrost dynamics.

[Figure]

Figure R1: Locations of the identified GPS stations in Northern Canada and the CALM sites and GTN-P boreholes. The IDs of GPS stations are presented. The types of permafrost distribution are indicated by various colours.

The following questions should be answered:

1. According to the description in the manuscript, the ground of the GPS is buried deep in the permafrost layer and will not change with the settlement of the active layer. The settlement results obtained are also sub-centimeter level, but the movement of the Earth's plate is vertical to the GPS receiver. The effect of displacement cannot be ignored. It is necessary to consider whether the spatial coordinates of the GPS antenna are constant, so that the ground subsidence can be measured with the GPS antenna as a reference.

**Response**: GPS-IR measurements obtained at the identified CACS sites are free of soil earth movement and glacier isostatic movement. Permafrost is the ground whose temperature is at or below 0 °C for at least two consecutive years. On top of permafrost is the active layer,

which thaws/freezes seasonally and accordingly is subject to subside/uplift (*Lines 33–35*). The monuments of the used CACS stations are anchored deep into permafrost, which means the monuments are stable with respect to the permafrost. Tectonic movements would have same impact on the monument foundation, the antenna aligned to the monument, and the active layer. The changes of reflector height, the distance between the antenna and surface, reflect directly the changes of the active layer and permafrost (*Lines 56–59*).

2. The photos of the stations showed the surrounding environment are still relatively complicated. The reflected SNR oscillation obtained by this environment should be disordered. The measured reflection SNR sequence diagrams and spectrum analysis results of several stations should be given.

**Response**: The photos are the only ones we can find. The azimuth coverages of the ground in the photos might not coincide with the ones we determine by following the method in section 2.2.

We show examples of SNR series in the determined azimuth coverages and their frequency spectrums of the 5 identified CACS sites in Fig. R2. We can clearly observe the consistent sinusoidal oscillations of SNR series. Such oscillations indicate that the surface condition within the determined azimuth coverage meets the requirements of GPS-IR. These SNR series are useful to retrieve reflector heights by frequency-spectrum analysis. In the right panels in Fig. R2, dominant frequencies, corresponding to reflector heights, of SNR series can be identified and they are clearly aligned.

Superimposed on the sinusoidal oscillations are high-frequency noises, which possibly caused by surface roughness and other unexpected disturbances. Those noises might introduce uncertainties to reflector height retrieved by spectrum analysis. To lower the uncertainty, more than 10 usable SNR series are required (*Lines 132–133*).

[Figure]

Figure R2: Examples of SNR series (left) and their frequency spectrums (right, expressed as reflector height in the x-axis) at the identified CACS site. The 2-order polynomial fits of SNR series have been removed. And the frequencies have been converted to reflector heights by equation (3) in Sect. 2. Each vertical red line in the right panels marks the dominant reflector height.

3. The noise of SNR measurement is relatively large, which results in the accuracy of snow thickness measurement with approximate specular reflection being only 0-5cm. The surface of the bare soil is rougher, and the error of height measurement will be larger, but the result is sub-centimeter, how to explain it.

**Response**: The uncertainties of our GPS-IR measurements are on the order of a few centimeters, which we demonstrate explicitly in lines 260–261. We also have published our results in PANGAEA (https://doi.pangaea.de/10.1594/PANGAEA.904347). The uncertainty is presented by the standard deviation of the mean value, i.e., the standard deviation divided by the square root of the sample size. Therefore, the magnitude of uncertainty partly depends on the sample size, which is more than 10 in our study.

4. The researches of Chew and Small (2014, 2016) showed that vegetation will affect the reflection signal. How to process the influence of vegetation on the reflection signal and height measurement needs to be explained in the manuscript.

**Response**: The study sites are located in the Canadian Arctic, where the biomes are dominantly Polar desert and tundra. The vegetation is either sparse or short. They are nearly transparent for L-band GPS signals. Therefore, the impact of vegetation on reflected signals and SNR is limited and can be ignored.

Reference:

Gruber, S. (2019). Ground subsidence and heave over permafrost: hourly time series reveal inter-annual, seasonal and shorter-term movement caused by freezing, thawing and water movement. The Cryosphere Discussions, 2019, 1–18. https://doi.org/10.5194/tc-2019-227

Hu, Y., Liu, L., Larson, K. M., Schaefer, K. M., Zhang, J., & Yao, Y. (2018). GPS Interferometric Reflectometry Reveals Cyclic Elevation Changes in Thaw and Freezing Seasons in a Permafrost Area (Barrow, Alaska). Geophysical Research Letters, 45(11), 5581–5589. https://doi.org/10.1029/2018GL077960

Liu, L., & Larson, M. (2018). Decadal changes of surface elevation over permafrost area estimated using reflected GPS signals. The Cryosphere, 12(2), 477–489. https://doi.org/10.5194/tc-12-477-2018

---

## Author Comment (AC3) · 22 Jan 2020

**Responses to SC1**

We thank Dr. Felipe G. Nievinski for his insightful and constructive comments. We have addressed all of them in the revised version of our manuscript. Our point-by-point replies are given below. As the reviewer may not be able to read our revised manuscript at this stage of The Cryosphere's review process, the line numbers refer to the previously submitted discussion paper, aiming to point out where the discussion paper has partly addressed the reviewer's comments.

Major:

1. The GPS antenna foundation should be discussed in more detail. Currently only the distinction between buildings and steel pipes is given. But not all steel pipes are the same. At least the foundation depth should be given. For example, ALRT is 6-m deep while REPL seems to have concrete slab under the metal pedestal.

**Response**: The monuments of the identified CACS stations are anchored into bedrocks. The monument materials are aluminium for REPL and IQAL, and steel for ALRT, RESO, and BAKE. The foundation depths of ALRT, RESO, REPL, and IQAL are 6 m, 3 m, 1.5 m, and 1 m, respectively. The foundation depth for BAKE is not available.

We have updated the Table 1 in the revised manuscript to explicit the foundation types and depths and monument materials.

Moderate:

2. The pioneering work of Hu et al. (2018) should be compared and contrasted with the present submission.

**Response**: Hu et al. (2018) proposed a composite model to fit surface elevation changes in both thawing and freezing seasons, by using the GPS-IR measurements in Barrow, Alaska from Liu and Larson (2018). Based on the measurements during 2004–2015, Liu and Larson observed a subsidence trend of $0.26 \pm 0.02$ cm yr-1, which indicated permafrost degradation. During the

same time span, the thaw season had a warming trend with 4.79 °C·day yr-1. This is consistent with our finding that the trend of surface elevation changes is negatively correlated with that of thawing indices (*Lines 251–252*). Air temperature is the dominant driver of permafrost dynamics at the study sites. The GPS site SG27 in Barrow and our newly identified sites in Northern Canada provides complementary study sites for permafrost studies. They can be used to study the permafrost changes across a broad region.

3. More details of the GPS-IR data processing is necessary. For example, what GPS signal was employed – L1-C/A?

**Response**: We use SNR data of GPS L1 C/A signals. In practice, we divide SNR series into individual parts corresponding to rising/setting satellite tracks. Then we remove the 2-order polynomial fits from them and use the residual ones. We conduct Lomb-Scargle Periodogram (LSP) analysis on any given SNR series to obtain its frequency spectrum, then use its peak value to represent the oscillating frequency, which is converted to reflector height. The oversampling parameter of LSP is determined to produce a resolution of 1 mm of reflector height. We use the software tools of GNSS Interferometric Reflectometry (Roesler and Larson, 2018).

We have revised the methodology section to show the details of data processing.

4. Authors should document the GPS receiver and antenna models used in each station, including the time of replacement, at least as supplementary material.

**Response**: We summarize the instrumentation information in Table R1, where the receiver types and antenna models during the data time span of each site are presented.

Table R1. Receiver and antenna types of the identified GPS stations.

| ID | Receiver type | Antenna model | Radome | Data time span | Source |
|---|---|---|---|---|---|
| ALRT | ASHTECH UZ-12 | ASH701945D_M | NONE | 2012–2017 | https://webapp.geod.nrcan.gc.ca/geod/datadonnees/cacsscca.php?locale=en |
| RESO | ASHTECH UZ-12 | ASH700936A_M | NONE | 2003–2014 | |

| | | | | | |
|---|---|---|---|---|---|
| REPL | TRIMBLE NETR9 | TRM59800.00 | NONE | 2014–2017 | |
| BAKE | TPS NETG3 (before 2013/07/11) TPS NET-G3A | TPSCR.G3 | NONE | 2010–2017 | |
| IQAL | TPS NETG3 (before 2011/09/12) TPS NET-G3A | TPSCR.G3 | NONE | 2010–2017 | |
| PONC | NOVATEL GSV4004 | NOV702GG | NONE | 2008–2018 | RINEX observation files |
| HALC | NOVATEL GSV4004 | NOV702GG | NONE | 2008– 2018 | |
| IQAC | NOVATEL GSV4004 | NOV702GG | NONE | 2008–2018 | |
| RANC | SEPT POLARXS | POLANT+_GG | NONE | 2014–2018 | |
| FSIC | SEPT POLARXS | POLANT+_GG | NONE | 2014–2018 | |
| FSMC | SEPT POLARXS | POLANT+_GG | NONE | 2014–2018 | |
| SANC | NOVATEL GSV4004 | NOV702GG | NONE | 2008–2018 | |

We have included the table R1 into the revised manuscript as supplementary.

5. Authors should also acknowledge some of the possible error sources in the simplistic mathematical formulation of eq.(1), where the phase term (phi) is not necessarily constant and can actually vary with soil moisture, vegetation cover, etc.

**Response**: We agree that equation (1), i.e., $SNR = A(e)\sin\left(\frac{4\pi H}{\lambda}\sin e + \phi\right)$ is the simplified expression of SNR series. Phase $\phi$ is not constant for each point of the series and also a function of satellite elevation angle. Equation (1) can be rewritten as $SNR = A(e)\sin\left(\frac{4\pi}{\lambda}\left(H + \frac{\lambda\phi}{4\pi\sin e}\right)\sin e\right)$. $\phi$ can be expressed as a part of reflector height as $\frac{\lambda\phi}{4\pi\sin e}$. Phases at different elevation angles have different impacts on reflector height. Taking phase as a constant in practice might introduce bias to reflector height. However, such bias is limited. Based on the simulations of Zavorotny et al. (2010), the variation range of the impact of phase on reflector height during the elevation angles 5°–30° is ~1 cm. Furthermore, we focus on the temporal variations of reflector height. So, the bias caused by taking phase as a constant does not have a significant impact on our results.

Regarding the vegetation, at the study sites, the biomes are Arctic desert or tundra. The ground is barely or sparsely vegetated. The vegetation is short enough, i.e., less than the wavelength of GPS signals. The vegetation is approximately transparent for GPS signals. The impact of vegetation on GPS-IR measurements is negligible.

Soil moisture is highly correlated to phases of SNR observations, manifested as an approximately linear relationship at low elevation angels (Larson et al., 2008; Zavorotny et al., 2010; and Chew et al., 2014). Changes of soil moisture introduce biases to reflector height retrievals. Soil moisture variation leads to surface permittivity changes, which affect reflected GPS signals and then phases of SNR observations. Such impact on phase at different elevation angles is different. The inconsistency of phase changes introduces bias to reflector height. Such impact is called compositional reflector height, as it manifests itself by a part of reflector height (Nievinski, 2013). Liu and Larson (2018) simulated the compositional height due to soil moisture changes between 15% and 40% and found that they are less than 2 cm and their variation range is less than 1 cm. In this study, the compositional heights and their variation range are expected to be limited, as the precipitation is light and limited due to the cold and dry

polar climate. Moreover, as we focus on the temporal variations of reflector heights at interannual and sub-decadal time scales, we expect a negligible impact of compositional heights on our results and interpretation.

Minor:

6. "The uncertainty of H-bar is represented by its standard deviation." -> It should be the standard error of the mean, i.e., the standard deviation divided by the square-root of the sample size.

**Response:** We have revised the manuscript to explicit that the uncertainty of the daily reflector height measurement is represented by the standard deviation of the mean value, i.e., the standard deviation divided by the square root of the sample size.

7. replace bullet for dot in: °CâAˇcˊday yr-1 -> °CÂuˑdayÂuˑyr-1

**Response:** We have revised the units of thawing index to °C·day.

8. for a 2-m-height antenna -> for a 2-m-high antenna [or: for 2-m antenna height.]

**Response:** We have revised the manuscript accordingly.

9. As the monument is deep anchored (e.g. Fig. 2), the GPS antenna is stable with respect to the permafrost -> If the monument is deep anchored (e.g. Fig. 2), the GPS antenna is stable with respect to the permafrost

**Response:** We have revised the manuscript accordingly.

Reference:

Chew, C. C., Small, E. E., Larson, K. M., & Zavorotny, V. U. (2014). Effects of near-surface soil moisture on GPS SNR data: Development of a retrieval algorithm for soil moisture. IEEE Transactions on Geoscience and Remote Sensing, 52(1), 537–543. https://doi.org/10.1109/TGRS.2013.2242332

Hu, Y., Liu, L., Larson, K. M., Schaefer, K. M., Zhang, J., & Yao, Y. (2018). GPS Interferometric Reflectometry Reveals Cyclic Elevation Changes in Thaw and Freezing Seasons in a Permafrost Area (Barrow, Alaska). Geophysical Research Letters, 45(11), 5581–5589. https://doi.org/10.1029/2018GL077960

Larson, K. M., Small, E. E., Gutmann, E. D., Bilich, A. L., Braun, J. J., & Zavorotny, V. U. (2008). Use of GPS receivers as a soil moisture network for water cycle studies. Geophysical Research Letters, 35(24), 1–5. https://doi.org/10.1029/2008GL036013

Liu, L., & Larson, M. (2018). Decadal changes of surface elevation over permafrost area estimated using reflected GPS signals. The Cryosphere, 12(2), 477–489. https://doi.org/10.5194/tc-12-477-2018

Nievinski, F. G.: Forward and inverse modeling of GPS multipath for snow monitoring, PhD thesis, University of Colorado, Boul- der, CO, USA, 2013.

Roesler, C., & Larson, K. M. (2018). Software tools for GNSS interferometric reflectometry ( GNSS-IR ). GPS Solutions, 22(3), 80. https://doi.org/10.1007/s10291-018-0744-8

Zavorotny, V. U., Larson, K. M., Braun, J. J., Small, E. E., Gutmann, E. D., & Bilich, A. L. (2010). A Physical Model for GPS Multipath Caused by Land Reflections: Toward Bare Soil Moisture Retrievals. IEEE Journal of Selected Topics in Applied Earth Observations and Remote Sensing, 3(1), 100–110. https://doi.org/10.1109/JSTARS.2009.2033608

---

## Author Response (AR1)

**Response to Editor**

We thank the editor and the reviewers for their insightful and constructive comments. We have addressed all of them and made the suggested changes in the new version of our manuscript. In addition, we took this chance to add the recent data and extend the data time span to 2018 in Alert and to 2019 in Repulse Bay and Iqaluit, and updated the results and interpretation correspondingly.

Please refer to the attached pdf for our point-by-point responses (in black) to the critical comments (in blue). Please note that the page/line numbers in our responses refer to the new line numbers in the revised manuscript.

The two anonymous referees and Dr. Felipe Nievinski mostly focused on the innovative aspect of the work from the perspective of methodology and raised many questions which substantially improve the manuscript. In addition to assessing the usability of existing GPS stations in Canada for studying permafrost, we also made significant efforts to interpret the GPS-IR results of surface elevation changes at the identified sites in the Canadian Arctic. We found surface subsidence in Alert (2010–2018), Resolute Bay (2003–2014), and Repulse Bay (2014–2019) with warming summers, which indicate permafrost degradation at these sites and air temperature is the dominant driver. We also found that, in Resolute Bay, ground uplift in the middle of summer in most of the years during 2003–2014 and end-of-thaw elevations were low with cool summers in 2013 and 2014. Such abnormal and complex changes cannot be explained by the simple Stefan equation. Our results and interpretation provide new insights into studying permafrost dynamics.

Our study is the first one of this kind assessing the usability of existing GPS stations in Canada for studying permafrost by GPS-IR. It is also the first one using multiple GPS sites to study frozen ground dynamics. This study is significant in the following three aspects: (1) it provides multiple usable GPS sites, complementary to the existing permafrost monitoring programs; (2) it provides daily measurements of surface elevation changes at five sites in the Canadian Arctic spanning from several years to more than a decade, with comprehensive consideration of possible errors; (3) it also puts new insights into permafrost studies. We believe that this study is worthwhile to be published in *The Cryosphere*. It will bring new and significant measurements and scientific findings to the readers of TC from permafrost and the related fields such as hydrology and remote sensing.

**Responses to RC1**

We thank the reviewer for his/her insightful and constructive comments. We have addressed all of them and made the suggested changes in the new version of our manuscript. In addition, we took this chance to add the recent data and extend the data time span to 2018 in Alert and to 2019 in Repulse Bay and Iqaluit, and updated the results and interpretation correspondingly. Below are our point-by-point responses (in black) to the critical comments (in blue). Please note that the page/line numbers in our responses refer to the new line numbers in the revised manuscript.

1. Please highlight what is really new and what is the outcome and applicability of this approach in a more prominent way.

**Response**: This comment asks about (1) innovative aspects, (2) outcome and applicability of this approach. Below, we are addressing the innovative aspects of this study in terms of methodology, new findings, and merits of using GPS-IR for studying permafrost. We will respond to the 2$^{nd}$ part of reviewer's comment in a separate thread.

(a) Methodology innovations:

(i) We implemented a framework to identify usable GPS stations from the existing networks to study permafrost by GPS-IR (*Page 4, Lines 121–144*). To our knowledge, this framework is the first of this kind for studying permafrost. Previous works by Liu and Larson (2018) and Hu et al. (2018) only used one GPS station (SG27 in Barrow, Alaska). Before our work, it is unknown how many among the 200+ GPS stations that are operating continuously in the circumpolar permafrost areas can be used for GPS-IR. The framework described in this work also serves as a reference for other researchers who would like to screen other GPS stations in other permafrost areas. To our knowledge, the only similar 'data mining' effort for the cryosphere is the PBO H2O project, which lists sites suitable for GPS-IR-based estimates of snow depth (and soil moisture). However, the PBO H2O project ended in 2017 and only archived products are available (https://cires1.colorado.edu/portal/).

(ii) Towards a robust use of GPS-IR, we summarized the limitations of using GPS-IR-estimated surface elevation changes in permafrost studies in section 5.4 (*Page 11, Lines 322–341*). These limitations are related to surrounding environments around stations, instrumentation maintenance, and auxiliary observations such as ground temperature, soil moisture, and ground ice. These limitations indicate that better location choice and maintenance of GPS sites are needed.

(iii) We considered comprehensively the error sources of GPS-IR-measured surface elevation changes, including tropospheric delay of GPS signals, antenna gain pattern, monument thermal extension/contraction, soil moisture, and vegetation in section 5.3 (*Page 10, Lines 280–320*). We evaluated the magnitudes of the biases caused by these variables and corresponding impacts on our results and interpretation.

(iv) We compared the advantages and limitations of GPS-IR measurements and InSAR observations, the latter of which has emerged as a tool for measuring elevation changes over permafrost areas. Our comparisons, detailed in section 5.5 and summarized in Table 4, clearly show that these two methods and their measurements are complementary to each other (*Page 11, Lines 342–356*). GPS-IR measurements can be used to calibrate and validate InSAR observations and provide baseline information for historical, current, and future remote sensing measurements from air and space.

(b) new findings:

(i) This study is the first one to use multiple GPS sites to study permafrost changes across a broad region by GPS-IR. It is also the first in Canada to use GPS-IR to study permafrost.

(ii) We observed that the surface subsided in Alert during 2012–2018, Resolute Bay during 2003–2014, and Repulse Bay during 2014–2019 (*Page 7, Lines 203–207*. We found the negative correlation between the linear trends of surface deformation and those of thawing indices (*Page 9, Lines 264–273*). It indicates that near-surface permafrost is sensitive to air temperature changes and that air temperature is the dominant driver for permafrost stability at these study sites.

(iii) In Resolute Bay, we found that end-of-thaw elevations during 2003–2012 had high negative correlation with the square root of thawing indices (*Page 8, Lines 238–242*). However, in 2013 and 2014, the end-of-thaw elevations were low with cool summers (*Page 9, Lines 251–252*). This phenomenon is possibly due to the Markovian behaviour of the active layer, which is worthwhile to be documented and investigated further.

(iv) In Resolute Bay, we observed summer heave of surface in most of the thaw seasons during 2003–2014 (*Page 7, Lines 212–213*). A similar phenomenon is also observed at a different site near Yellowknife in Canada by Gruber (2019) using an inclinometer. These findings reveal that frozen dynamics is rather complex and cannot be all explained by simple Stefan equations.

(c) Merits of using GPS-IR for studying permafrost:

The identified usable GPS sites in this study complement the existing permafrost monitoring programs such as CALM (circumpolar active layer monitoring) and GTN-P (global terrestrial network for permafrost), and provide multi-year, continuous, and daily measurements with intermediate spatial coverages. The changes of permafrost areas have large spatial heterogeneity, due to location, topography, precipitation, and vegetation. Despite the significant increase in the number of in situ sites in the past decades, the CALM and GTN-P sites are still sparse and unevenly distributed. The 12 suitable GPS stations in our study distributed across Northern Canada. Their locations fill in the spatial gaps of the CALM and GTN-P sites (Fig. R1). Moreover, the spatial coverage of GPS-IR at one site is on the order of 1000 m$^2$, nicely bridging point observations and regional-scale remote sensing measurements (*Page 2, Lines 54–56*). Moreover, numerous GPS stations, e.g., RESO in resolute Bay, have been in operation for more than a decade. They can provide long-term, continuous, and daily measurements, which aid in studying permafrost in a detailed manner and provide new insights to permafrost dynamics.

We have refined the abstract (*Page 1, Lines 19–22*) and the conclusion (*Page 12, Lines 358–381*) to explicitly show our innovations.

[Figure]

Figure R1: Locations of the identified GPS stations in Northern Canada and the CALM sites and GTN-P boreholes. The IDs of GPS stations are presented. The types of permafrost distribution are indicated by various colors.

1. Please highlight what is really new and what is the outcome and applicability of this approach in a more prominent way.

**Response**: Corresponding to the innovations presented above, the major outcomes of this approach in this study are (1) a framework guiding to identify suitable GPS stations in permafrost areas, where GPS-IR is feasible, (2) 12 identified suitable GPS stations distributed across Northern Canada, (3) Multi-year, continuous, and daily measurements of surface deformation at five geodetic-quality GPS sites, (4) new findings in the dynamics of the active layer and permafrost.

Applicability of GPS-IR depends on the surface conditions, which should be relatively horizontal and smooth. This is the reason why we proposed a framework to identify suitable ones from existing stations. We also summarize the limitations of GPS-IR measurements in permafrost studies. The

applicability and limitations of GPS-IR indicate that better location choices and instrument maintenance should be made in the future to fully realize the potentials of the GPS stations installed in frozen areas.

2. It is already proved by Larson et al. (2008) that the reflector height (H) and the phase of SNR observations are highly correlated. Therefore, some of the H variations should come from the phase variations.

**Response**: We do not think Larson et al. (2008) proved the high correlation between reflector height and the phase of SNR series. Instead, their study found a high correlation between the phase of SNR series and surface soil moisture. To explain this finding, they introduced the term of apparent reflector height, which is converted from the phase (equation (1), i.e., $SNR = A(e)\sin\left(\frac{4\pi H}{\lambda}\sin e + \phi\right)$, can be rewritten as $SNR = A(e)\sin(\frac{4\pi}{\lambda}\left(H + \frac{\lambda\phi}{4\pi\sin e}\right)\sin e))$. They thought that soil moisture affects apparent reflector height (i.e., "When the soil is wet, the apparent reflector is close to the surface; as it dries, the reflection depth is several cm deeper" in Larson et al. (2008)), then affects phase. Apparent reflector height (converted from the phase) cannot be mixed up with the reflector height (converted from the frequency).

3. SNR observations also are a function of GPS receiving antenna and GPS signals. The authors have not discussed what types of antennas and signals are investigated, and how their impacts have been moved from the estimated heights.

**Response**: In this study, we used SNR data of GPS L1 C/A signals, as L1 C/A is the legacy civilian signal broadcasted by all satellites and long-term measurements can be obtained. As for the antenna, its gain pattern impacts GPS-IR measurements. The GPS stations used in this study were originally installed and maintained for geodetic or ionospheric studies. Their antennas were designed to favor direct signals with high elevation angles and suppressing signals with low and negative elevation angles, by using asymmetric antenna gain patterns. During the data time spans, antennas were not replaced. The impact of antenna gain pattern can be regarded as a system bias, and barely impact the GPS-IR results.

In the revised manuscript, we have explicitly presented the antenna types in the supplementary by Table S1 and included more details of data processing (*Page 4, Lines 106–112; Page 5, Lines 146–148*). And we also have added a subsection (*section 5.3, Page 10, Lines 280–320*) discussing possible error sources of GPS-IR measurements, including antenna gain patterns.

4. In addition, the penetration depth of microwave signals should be physically estimated to correct the estimated heights.

**Response**: The penetration depth of microwave signals, i.e., the depth where the power of signal reduces to $1/e$ of its value at soil surface, is not appropriate for defining the depth of reflector (Chew et al., 2014). The signal might penetrate into soil to some depth, but it does not mean that a significant part of the signal would reflect back from that depth to antenna. The interaction between signal and ground surface is rather complex. Comparing to penetration depth, it is more important to consider how changes of ground permittivity affect SNR data. Zavorotny et al. (2010) built a forward model to simulate SNR, with consideration of antenna gain pattern and surface reflectivity but not penetration depth. The simulated SNR and their GPS-IR retrievals have a good agreement with the experimental ones. This simulation indicates that reflected signal from the penetration depth has a minor contribution to SNR. Furthermore, we focus on the temporal variations of reflector heights. We do not expect significant temporal changes in penetration depth during the data time spans. Therefore, we do not see a need to estimate penetration depth and correct for it.

5. The authors just used low elevation angle observations as SNR oscillations are clearer. However, the tropospheric impact is not negligible for low elevation angle observations even for 2-3 m antenna heights, so its impact should be studied in the paper. The tropospheric refractions seem to have a seasonal impact on GPS-IR (Williams and Nievinski, 2017). The authors have not corrected their GPS-IR solutions. Therefore, the reported slopes are tropospheric-contaminated.

**Response**: Tropospheric refraction may introduce biases in the estimated reflector height, which, however, barely affect the retrieved temporal elevation changes as presented in this study. The tropospheric bias mainly depends on antenna height and atmospheric conditions at a given satellite elevation angle (Williams and Nievinski, 2017). The GPS stations used in this study are located in the Canadian Arctic, where the climate is dry and cold. The antenna heights of these stations are ~2 m. Conceptually, the tropospheric bias of GPS-IR measurements of reflector height is limited.

We calculate the tropospheric bias by using the astronomical refraction model of Bennett (1982) and SNR data in Resolute Bay in the thaw season of 2014. The results show that the mean bias is 1.6 cm, and that they are relatively steady with a variation range of merely 3 mm (Fig. R2(a)). The biases do not show a significant seasonal variability. And their magnitudes are comparable to the uncertainties of our GPS-IR measurements (Fig. R2(b)). As we focus on the temporal variations of reflector heights, instead of their absolute values, it is unnecessary to correct for the tropospheric biases.

We have added a subsection (*section 5.3, Page 10, Lines 280–320*) discussing possible error sources of GPS-IR measurements, including tropospheric delays.

[Figure]

Figure R2: (a) Time series of tropospheric biases in estimated reflector heights in Resolute Bay in the thaw season (i.e., DOY 192–250) of 2014. They are the mean values of the tropospheric biases of all satellite tracks; their standard deviations are indicated by error bars. (b) Time series of GPS-IR-estimated surface elevation changes shown in Fig. 6 in the revised manuscript and their 7-day moving averages.

6. In addition, the authors used different antenna monuments (Figure 4), but they did not study the thermal expansion of monuments.

**Response**: The monuments of the 5 identified CACS stations are made of aluminium or Galvanized/Stainless steel, whose linear thermal expansion/contraction coefficients are $11\sim13 \times 10^{-6}$ and $23.1 \times 10^{-6}$ m/(m K), respectively. Given a temperature variation range of 20 °C in thaw season, for a 2-m-high aluminum/steel monument, the magnitude of thermal expansion is less than 1 mm, at least one order of magnitude smaller than the elevation changes. The thermal expansion/contraction impact is

ignorable for GPS-IR measurements. We have added a subsection (*section 5.3, Page 10, Lines 280–320*) discussing possible error sources of GPS-IR measurements, including monument material.

7. The authors used the mean value of each year after the outliers are rejected to estimate and report the slopes, while the results presented in Figure 5 are somehow confusing. For example, at Alert what is the mean for 2016? It seems it is very different from the other 5 years and 2016 cannot represent the reported slope. There are other examples like 2014 at Iqaluit, 2004 and 2011 at Resolute Bay.

**Response**: Sorry about the confusion caused. We would like to clarify that we use all daily GPS-IR measurements in thaw seasons during the entire data time span to obtain the linear trend at each site. For example, in Alert, the linear trend corresponds to the daily measurements in thaw seasons during 2012–2018. We do not use the mean value of each year or reject any outliers.

The deviation of the measurements of Alert in 2016 (or those of Iqaluit in 2014 and those of Resolute Bay in 2004 and 2011) from the best linear fit is due to the interannual variability of changes in the active layer and near-surface permafrost. Such interannual variability is related to environmental variables including precipitation, soil temperature, soil moisture content, and ground ice condition. However, it is challenging to study the interannual variability as most of these ancillary records are not available. This is one of the limitations of using GPS-IR measurements to study permafrost as we discussed in section 5.4 (*Page 11, Lines 322–341*).

8. In addition, it is more complex at Bakes Lake as it seems just a linear fit doesn't represent the GPS station and higher-order polynomials should be used. It would be easier if the median reflector heights for each year were also plotted together with the time series in Figure 5. That would help us to understand if the linear fit is good enough to report surface elevation changes.

**Response**: The linear trend is superimposed by seasonal variation, interannual variability, and possibly sub-decadal pattern. Given the data time span, it is not justified to simply use a high-order polynomial to fit the time series, as it is not long enough. The GPS site in Baker Lake is still operating. It would be clearer whether a sub-decadal pattern exists when we have at least decade-long measurements.

We tend not to use the median reflector height of each year. The time series of measurements is a combination of linear trend, interannual variability, seasonal variation, and possible sub-decadal pattern. It cannot be represented by the time series of median reflector heights.

**Responses to RC2**

We thank the reviewer for his/her insightful and constructive comments. We have addressed all of them and made the suggested changes in the new version of our manuscript. In addition, we took this chance to add the recent data and extend the data time span to 2018 in Alert and to 2019 in Repulse Bay and Iqaluit, and updated the results and interpretation correspondingly. Below are our point-by-point responses (in black) to the critical comments (in blue). Please note that the page/line numbers in our responses refer to the new line numbers.

GPS-IR has been used in many terrestrial parameter inverses. The determination of permafrost surface elevation changes via GPS-IR is an interest research. The theory and method of measuring the elevation is the universal method in snow depth and water elevation inverses proposed by Larson. The main work of the manuscript is to filter out a large number of stations that meet the conditions and select the GPS data that can be used in elevation measure. So, this manuscript is unacceptable in its present form and need to add more innovative work.

**Response**: The main work is not limited to the 12 identified useful GPS stations. Below, we present the innovative aspects of this study in terms of methodology, new findings, and merits of using GPS-IR for studying permafrost.

(a) Methodology innovations:

(i) We implemented a framework to identify usable GPS stations from the existing networks to study permafrost by GPS-IR (*Page 4, Lines 121–144*). To our knowledge, this framework is the first of this kind for studying permafrost. Previous works by Liu and Larson (2018) and Hu et al. (2018) only used one GPS station (SG27 in Barrow, Alaska). Before our work, it is unknown how many among the 200+ GPS stations that are operating continuously in the circumpolar permafrost areas can be used for GPS-IR. The framework described in this work also serves as a reference for other researchers who would like to screen other GPS stations in other permafrost areas. To our knowledge, the only similar 'data mining' effort for the cryosphere is the PBO H2O project, which lists sites suitable for GPS-IR-based estimates of snow depth (and soil moisture). However, the PBO H2O project ended in 2017 and only archived products are available (https://cires1.colorado.edu/portal/).

(ii) Towards a robust use of GPS-IR, we summarized the limitations of using GPS-IR-estimated surface elevation changes in permafrost studies in section 5.4 (*Page 11, Lines 322–341*). These limitations are related to surrounding environments around stations, instrumentation maintenance, and auxiliary

observations such as ground temperature, soil moisture, and ground ice. These limitations indicate that better location choice and maintenance of GPS sites are needed.

(iii) We considered comprehensively the error sources of GPS-IR-measured surface elevation changes, including tropospheric delay of GPS signals, antenna gain pattern, monument thermal extension/contraction, soil moisture, and vegetation in section 5.3 (*Page 10, Lines 280–320*). We evaluated the magnitudes of the biases caused by these variables and corresponding impacts on our results and interpretation.

(iv) We compared the advantages and limitations of GPS-IR measurements and InSAR observations, the latter of which has emerged as a tool for measuring elevation changes over permafrost areas. Our comparisons, detailed in section 5.5 and summarized in Table 4, clearly show that these two methods and their measurements are complementary to each other (*Page 11, Lines 343–356*). GPS-IR measurements can be used to calibrate and validate InSAR observations and provide baseline information for historical, current, and future remote sensing measurements from air and space.

(b) new findings:

(i) This study is the first one to use multiple GPS sites to study permafrost changes across a broad region by GPS-IR. It is also the first in Canada to use GPS-IR to study permafrost.

(ii) We observed that the surface subsided in Alert during 2012–2018, Resolute Bay during 2003–2014, and Repulse Bay during 2014–2019 (*Page 7, Lines 203–207*). We found the negative correlation between the linear trends of surface deformation and those of thawing indices (*Page 9, Lines 264–268*). It indicates that near-surface permafrost is sensitive to air temperature changes and that air temperature is the dominant driver for permafrost stability at these study sites.

(iii) In Resolute Bay, we found that end-of-thaw elevations during 2003–2012 had high negative correlation with the square root of thawing indices (*Page 8, Lines 238–242*). However, in 2013 and 2014, the end-of-thaw elevations were low with cool summers (*Page 9, Lines 251–252*). This phenomenon is possibly due to the Markovian behaviour of the active layer, which is worthwhile to be documented and investigated further.

(iv) In Resolute Bay, we observed summer heave of surface in most of the thaw seasons during 2003–2014 (*Page 7, Lines 212–213*). A similar phenomenon is also observed at a different site near

Yellowknife in Canada by Gruber (2019) using an inclinometer. These findings reveal that frozen dynamics is rather complex and cannot be all explained by simple Stefan equations.

(c) Merits of using GPS-IR for studying permafrost:

The identified usable GPS sites in this study complement the existing permafrost monitoring programs such as CALM (circumpolar active layer monitoring) and GTN-P (global terrestrial network for permafrost), and provide multi-year, continuous, and daily measurements with intermediate spatial coverages. The changes of permafrost areas have large spatial heterogeneity, due to location, topography, precipitation, and vegetation. Despite the significant increase in the number of in situ sites in the past decades, the CALM and GTN-P sites are still sparse and unevenly distributed. The 12 suitable GPS stations in our study distributed across Northern Canada. Their locations fill in the spatial gaps of the CALM and GTN-P sites (Fig. R1). Moreover, the spatial coverage of GPS-IR at one site is on the order of 1000 m$^2$, nicely bridging point observations and regional-scale remote sensing measurements (*Page 2, Lines 54–56*). Moreover, numerous GPS stations, e.g., RESO in resolute Bay, have been in operation for more than a decade. They can provide long-term, continuous, and daily measurements, which aid in studying permafrost in a detailed manner and provide new insights to permafrost dynamics.

We have refined the abstract (*Page 1, Lines 19–22*) and the conclusion (*Page 12, Lines 358–381*) to explicitly show our innovations.

[Figure]

Figure R1: Locations of the identified GPS stations in Northern Canada and the CALM sites and GTN-P boreholes. The IDs of GPS stations are presented. The types of permafrost distribution are indicated by various colors.

The following questions should be answered:

1. According to the description in the manuscript, the ground of the GPS is buried deep in the permafrost layer and will not change with the settlement of the active layer. The settlement results obtained are also sub-centimeter level, but the movement of the Earth's plate is vertical to the GPS receiver. The effect of displacement cannot be ignored. It is necessary to consider whether the spatial coordinates of the GPS antenna are constant, so that the ground subsidence can be measured with the GPS antenna as a reference.

**Response**: GPS-IR measurements obtained at the identified CACS sites are free of soil earth movement and glacier isostatic movement. Permafrost is the ground whose temperature is at or below 0 °C for at least two consecutive years. On top of permafrost is the active layer, which thaws/freezes seasonally

and accordingly is subject to subside/uplift (*Page 2, Lines 33–35*). The monuments of the used CACS stations are anchored deep into permafrost, which means the monuments are stable with respect to the permafrost. Tectonic movements would have the same impact on the monument foundation, the antenna aligned to the monument, and the active layer. The changes of reflector height, the distance between the antenna and surface, directly reflect the changes of the active layer and permafrost (*Page 2, Lines 55–58*).

2. The photos of the stations showed the surrounding environment are still relatively complicated. The reflected SNR oscillation obtained by this environment should be disordered. The measured reflection SNR sequence diagrams and spectrum analysis results of several stations should be given.

**Response**: The photos are the only ones we can find. The azimuth coverages of the ground in the photos might not coincide with the ones we determine by following the method in section 2.2.

We show examples of SNR series in the determined azimuth coverages and their frequency spectrums of the 5 identified CACS sites in Fig. R2. We can clearly observe the consistent sinusoidal oscillations of SNR series. Such oscillations indicate that the surface condition within the determined azimuth coverage meets the requirements of GPS-IR. These SNR series are useful to retrieve reflector heights by frequency-spectrum analysis. In the right panels in Fig. R2, dominant frequencies, corresponding to reflector heights, of SNR series can be identified and they are clearly aligned.

Superimposed on the sinusoidal oscillations are high-frequency noises, which possibly caused by surface roughness and other unexpected disturbances. Those noises might introduce uncertainties to reflector height retrieved by spectrum analysis. To lower the uncertainty, more than 10 usable SNR series are required (*Page 5, Lines 143–144*).

We have included Fig. R2 as supplementary.

[Figure]

Figure R2: Examples of SNR series (left) and their frequency spectrums (right, expressed as reflector height in the x-axis) at the identified CACS site. The 2-order polynomial fits of SNR series have been removed. And the frequencies have been converted to reflector heights by equation (3) in Sect. 2. Each vertical red line in the right panels marks the dominant reflector height.

3. The noise of SNR measurement is relatively large, which results in the accuracy of snow thickness measurement with approximate specular reflection being only 0-5cm. The surface of the bare soil is rougher, and the error of height measurement will be larger, but the result is sub-centimeter, how to explain it.

**Response**: The uncertainties of our GPS-IR measurements are on the order of a few centimeters, which we demonstrate explicitly in Line 322 (*Page* 11). We also have published our results in PANGAEA (https://doi.pangaea.de/10.1594/PANGAEA.904347). The uncertainty is presented by the standard deviation of the mean value, i.e., the standard deviation divided by the square root of the sample size. Therefore, the magnitude of uncertainty partly depends on the sample size, which is more than 10 in our study.

4. The researches of Chew and Small (2014, 2016) showed that vegetation will affect the reflection signal. How to process the influence of vegetation on the reflection signal and height measurement needs to be explained in the manuscript.

**Response**: The study sites are located in the Canadian Arctic, where the biomes are dominantly Polar desert and tundra. The vegetation is either sparse or short. They are nearly transparent for L-band GPS signals. Therefore, the impact of vegetation on reflected signals and SNR is limited and can be ignored. We have added a subsection (*section 5.3, Page 10, Lines 280–320*) discussing possible error sources of GPS-IR measurements, including vegetation.

**Response**: We summarize the instrumentation information in Table R1, where the receiver types and antenna models during the data time span of each site are presented. We have included the table R1 into the revised manuscript as supplementary.

Table R1. Receiver and antenna types of the identified GPS stations.

| ID | Receiver type | Antenna model | Radome | Data time span | Source |
|---|---|---|---|---|---|
| ALRT | ASHTECH UZ-12 | ASH701945D _M | NONE | 2012–2018 | https://webapp.geod. nrcan.gc.ca/geod/dat adonnees/cacsscca.p hp?locale=en |
| RESO | ASHTECH UZ-12 | ASH700936A _M | NONE | 2003–2014 | |
| REPL | TRIMBLE NETR9 | TRM59800.00 | NONE | 2014–2019 | |

| | | | | | |
|---|---|---|---|---|---|
| BAKE | TPS NETG3 (before 2013/07/11) TPS NET-G3A | TPSCR.G3 | NONE | 2010–2017 | |
| IQAL | TPS NETG3 (before 2011/09/12) TPS NET-G3A | TPSCR.G3 | NONE | 2010–2019 | |
| PONC | NOVATEL GSV4004 | NOV702GG | NONE | 2008–2018 | RINEX observation files |
| HALC | NOVATEL GSV4004 | NOV702GG | NONE | 2008– 2018 | |
| IQAC | NOVATEL GSV4004 | NOV702GG | NONE | 2008–2018 | |
| RANC | SEPT POLARXS | POLANT+_G G | NONE | 2014–2018 | |
| FSIC | SEPT POLARXS | POLANT+_G G | NONE | 2014–2018 | |
| FSMC | SEPT POLARXS | POLANT+_G G | NONE | 2014–2018 | |
| SANC | NOVATEL GSV4004 | NOV702GG | NONE | 2008–2018 | |

5. Authors should also acknowledge some of the possible error sources in the simplistic mathematical formulation of eq.(1), where the phase term (phi) is not necessarily constant and can actually vary with soil moisture, vegetation cover, etc.

**Response**: We agree that equation (1), i.e., $SNR = A(e)\sin\left(\frac{4\pi H}{\lambda}\sin e + \phi\right)$ is the simplified expression of SNR series. Phase $\phi$ is not constant for each point of the series and also a function of satellite elevation angle. Equation (1) can be rewritten as $SNR = A(e)\sin\left(\frac{4\pi}{\lambda}\left(H + \frac{\lambda\phi}{4\pi\sin e}\right)\sin e\right)$. $\phi$

can be expressed as a part of reflector height as $\frac{\lambda\phi}{4\pi\sin e}$. Phases at different elevation angles have different impacts on reflector height. Taking phase as a constant in practice might introduce bias to reflector height. However, such bias is limited. Based on the simulations of Zavorotny et al. (2010), the variation range of the impact of phase on reflector height during the elevation angles 5°–30° is ~1 cm. Furthermore, we focus on the temporal variations of reflector height. So, the bias caused by taking phase as a constant does not have a significant impact on our results.

Regarding the vegetation, at the study sites, the biomes are Arctic desert or tundra. The ground is barely or sparsely vegetated. The vegetation is short enough, i.e., less than the wavelength of GPS signals. The vegetation is approximately transparent for GPS signals. The impact of vegetation on GPS-IR measurements is negligible.

Soil moisture is highly correlated to phases of SNR observations, manifested as an approximately linear relationship at low elevation angles (Larson et al., 2008; Zavorotny et al., 2010; and Chew et al., 2014). Changes of soil moisture introduce biases to reflector height retrievals. Soil moisture variation leads to surface permittivity changes, which affect reflected GPS signals and then phases of SNR observations. Such impact on phase at different elevation angles is different. The inconsistency of phase changes introduces bias to reflector height. Such impact is called compositional reflector height, as it manifests itself by a part of reflector height (Nievinski, 2013). Liu and Larson (2018) simulated the compositional height due to soil moisture changes between 15% and 40% and found that they are less than 2 cm and their variation range is less than 1 cm. In this study, the compositional heights and their variation range are expected to be limited, as the precipitation is light and limited due to the cold and dry polar climate. Moreover, as we focus on the temporal variations of reflector heights at interannual and sub-decadal time scales, we expect a negligible impact of compositional heights on our results and interpretation.

We have added a subsection (*section 5.3, Page 10, Lines 280–320*) discussing possible error sources of GPS-IR measurements, including tropospheric delays, soil moisture, vegetation, antenna gain patterns, and monument materials.

Minor:

6. "The uncertainty of H-bar is represented by its standard deviation." -> It should be the standard error of the mean, i.e., the standard deviation divided by the square-root of the sample size.

**Response:** We have revised the manuscript to explicit that the uncertainty of the daily reflector height measurement is represented by the standard deviation of the mean value, i.e., the standard deviation divided by the square root of the sample size. (*Page 5, Lines 142–143*)

7. replace bullet for dot in: °CâAˇc´day ̊yr-1 -> °CÂu ̊dayÂu ̊yr-1

**Response:** We have revised the units of thawing index to °C·day. (e.g., *Page 9, Line 255*)

8. for a 2-m-height antenna -> for a 2-m-high antenna [or: for 2-m antenna height.]

**Response:** We have revised the manuscript accordingly. (*Page 2, Line 54*)

9. As the monument is deep anchored (e.g. Fig. 2), the GPS antenna is stable with respect to the permafrost -> If the monument is deep anchored (e.g. Fig. 2), the GPS antenna is stable with respect to the permafrost

**Response:** We have revised the manuscript accordingly. (*Page 4, Lines 114–115*)

[revised manuscript text omitted]

---

## Author Response (AR2)

**Response to Editor**

We thank the editor and reviewers for their constructive comments. We have addressed all of them and made the suggested changes in the new version of our manuscript. Please refer to the attached pdf for our point-by-point responses (in black) to the critical comments (in blue). Please note that the page/line numbers in our responses refer to the new line numbers in the revised manuscript.

1. Line 107: "2-order". Should it be "2nd order"?

**Response**: We have changed "2-order" to "2nd order" accordingly. (*Page 4, Line 107*)

2. Line 147:"over from several" should be "over several"

**Response**: We have revised the manuscript accordingly. (*Page 5, Line 147*)

3. Line 180: "bedrocks" should be "bedrock" (singular)

**Response**: We have changed "bedrocks" to "bedrock" accordingly. (*Page 6, Line 180*)

4. Line 180: "Figure 4". Should it be "Fig. 4" as this is the style used elsewhere in the document?

**Response**: Based on the manuscript preparation guidelines of TC, the abbreviation "Fig." is only used when it appears in running text. We use "Figure 4" as it comes at the beginning of the sentence.

5. Line 297: "soil moisture has slightly" should be "soil moisture has a slightly"

**Response**: We have revised the manuscript accordingly. (*Page 10, Line 297*)

6. Lines 306-307: "barely or sparsely vegetated". Should it be "bare or sparsely..."?

**Response**: We have changed "barely or sparsely vegetated" to "bare or sparsely vegetated". (*Page 10, Line 306*)

7. Line 430: D. K., Kale replace by Kale, D. K.

**Response**: The corresponding citation is during Lines 429–431. "D. K., Kale" are the first and last names of Milling, D. K. and Kale, Z. C., respectively.

[revised manuscript text omitted]